# Temperature increase drives critical slowing down of fish ecosystems

Jie Li[1], Matteo Convertino[2]*

**1** Nexus Group, Laboratory of Information Communication Networks, Graduate School of Information Science and Technology, Hokkaido University, Sapporo, Japan, **2** Institute of Environment and Ecology, Tsinghua Shenzhen International Graduate School, Tsinghua University, Shenzhen, China

\* matconv.uni@gmail.com

**Data Availability Statement:** All relevant data are within the manuscript and its Supporting information files.

## Abstract

Fish ecosystems perform ecological functions that are critically important for the sustainability of marine ecosystems, such as global food security and carbon stock. During the 21st century, significant global warming caused by climate change has created pressing challenges for fish ecosystems that threaten species existence and global ecosystem health. Here, we study a coastal fish community in Maizuru Bay, Japan, and investigate the relationships between fluctuations of ST, abundance-based species interactions and salient fish biodiversity. Observations show that a local 20% increase in temperature from 2002 to 2014 underpins a long-term reduction in fish diversity ($\sim$25%) played out by some native and invasive species (e.g. Chinese wrasse) becoming exceedingly abundant; this causes a large decay in commercially valuable species (e.g. Japanese anchovy) coupled to an increase in ecological productivity. The fish community is analyzed considering five temperature ranges to understand its atemporal seasonal sensitivity to ST changes, and long-term trends. An optimal information flow model is used to reconstruct species interaction networks that emerge as topologically different for distinct temperature ranges and species dynamics. Networks for low temperatures are more scale-free compared to ones for intermediate (15-20°C) temperatures in which the fish ecosystem experiences a first-order phase transition in interactions from locally stable to metastable and globally unstable for high temperatures states as suggested by abundance-spectrum transitions. The dynamic dominant eigenvalue of species interactions shows increasing instability for competitive species (spiking in summer due to intermediate-season critical transitions) leading to enhanced community variability and critical slowing down despite higher time-point resilience. Native competitive species whose abundance is distributed more exponentially have the highest total directed interactions and are keystone species (e.g. *Wrasse and Horse mackerel*) for the most salient links with cooperative decaying species. Competitive species, with higher eco-climatic memory and synchronization, are the most affected by temperature and play an important role in maintaining fish ecosystem stability via multitrophic cascades (via cooperative-competitive species imbalance), and as bioindicators of change. More climate-fitted species follow temperature increase causing larger divergence divergence between competitive and cooperative species. Decreasing dominant eigenvalues and lower relative network optimality for warmer oceans indicate fishery more attracted toward

**Funding:** M.C. and J.L. gratefully acknowledge the funding provided in the form of a grant by the GI-CORE Global Station for Big-Data and Cybersecurity at Hokkaido University, Sapporo, JP. M.C. acknowledges the Shenzhen High-quality Overseas Talents-Class A funding, and the FY2020 SOUSEI Support Program and Award for Young Researchers awarded in the form of a grant by the Executive Office for Research Strategy to the Top 20% scientists in terms of productivity and citations at Hokkaido University. M.C. also acknowledges the support provided by Microsoft AI for Earth in the form of computational resources. The funders had no role in study design, data collection and analysis, decision to publish, or preparation of the manuscript.

**Competing interests:** The authors have declared that no competing interests exist.

persistent oscillatory states, yet unpredictable, with lower cooperation, diversity and fish stock despite the increase in community abundance due to non-commercial and venomous species. We emphasize how changes in species interaction organization, primarily affected by temperature fluctuations, are the backbone of biodiversity dynamics and yet for functional diversity in contrast to taxonomic richness. Abundance and richness manifest gradual shifts while interactions show sudden shift. The work provides data-driven tools for analyzing and monitoring fish ecosystems under the pressure of global warming or other stressors. Abundance and interaction patterns derived by network-based analyses proved useful to assess ecosystem susceptibility and effective change, and formulate predictive dynamic information for science-based fishery policy aimed to maintain marine ecosystems stable and sustainable.

*"Oceani Sunt Servandi"* (old Roman incision)

# 1 Introduction

## 1.1 Impacts of ocean warming on marine fisheries

Fish species are a vital component of marine ecosystems and remain one of major sources of food and nutrition feeding hundreds of millions of people around the world. This is especially important for coastal undeveloped and developing countries, where fish provide inhabitants with as much as 50% of animal protein intake [1] and are a major economic income. According to annual reports from FAO, the global food fish consumption and average fish consumption per capita increased at an average rate of 3.1% from 1960s, and 1.5% per year, respectively [2]. Despite their critical role in maintaining the sustainability of marine ecosystems and global food security, fish ecosystems are all along under increasing anthropogenic pressure from pollution, habitat degradation and climate change that has been steadily warming the sea water. Marine fish ecosystems are prototypical complex systems that are highly dynamical and sensitive to external environmental factors including sea temperature (ST). Ocean warming is altering fish ecosystems with deep impacts on their ecology (behavior, biomass, range and abundance to name just a few) and environment (for instance on biogeochemical cycles) and hence with negative feedback on the climate itself. Data from the US National Oceanic and Atmospheric Administration (NOAA) show that ST anomaly increased from -0.02˚C in 1974 to 0.79˚C in 2016 which is the largest anomaly observed in last 100 years [3]. By comparing ST data from last two decades (1999–2019) to two decades before that (1979–1999), ocean warming had dramatically sped up in last two decades [3, 4]. "Fish are like Goldilocks: they don't like their water too hot or too cold, but just right.", said Malin L. Pinsky, a co-author of the study [5], where scientists found that some few species populations benefited from ocean warming, but more of them suffered. These results stress the fact that ocean warming is making fish species diversity and populations decline, yet threatening the sustainability of marine ecosystems and food supply for millions of people around the world. Therefore, it is imperative to unravel the complex relationships between marine ecosystems and anomalous changes of

ST caused by climate change, so that stakeholders can design optimal ecosystem restoration and policy to mitigate the negative effects of climate change on these ecosystems.

Due to increasing ecological and societal concerns about the sustainability of marine ecosystems –where fishery are following a rapid trend of depletion [6]—coupled to environmental degradation of habitats under ocean warming and development pressure, assessing the systemic vulnerability (mode, extent, location and species) of these ecosystems is considerably important. Fishes are particularly critical considering both their multitrophic (positive and negative) feedback on species (from microbes to humans; examples are emerging viruses in the oceans [7, 8]) and the environment (in terms of biogeochemical cycles affecting for instance algal blooms and carbon sequestration [9, 10]) and their high potential as multiscale biosensors of marine ecosystem health and change. On the one hand, most studies conduct these research fields only by visualizing taxonomic and abundance time series and observing the variations of species biodiversity, distribution and abundance at temporal scale. To identify possible environmental factors responsible for the deterioration of marine fish ecosystem, linear and non-linear relationships between taxonomic and abundance indicators and climate change are studied by comparing the observations to oceanographic climatic data. For example, [11] found that increasingly warming ST altered the composition of marine fish catches, with an increase in the proportion of warmer water species catches at higher latitudes and a decrease of subtropical species catches at lower latitudes by plotting sea surface temperature in the past four decades and calculating mean temperature as an indicator to describe the preference temperature of fish species. [5] applied temperature-dependent population models to measure negative effects of ocean warming on the productivity of 235 populations of 124 species in 38 ecological regions by analyzing the data of marine fishery production from 1930 to 2010 and computed maximum sustainable yield over time. [12] plotted time-series abundance and temperature collected in the North Sea since 1960s and west of Scotland since 1980s, and estimated the trend of length at age and growth parameters including absolute growth rate over time and confirmed the fact that the growth and maturation of fish species correlated to ST, and the mean length of herring populations steadily declined across multiple age groups. [13] examined influences of ST on global biomass transfers from marine secondary production to fish stocks, and predicted that trophic transfer efficiency (TTE) and biomass residence time (BRT) would decrease until 2040 and remain relatively stable after 2040, also by plotting the past observations of the trend of TTE and BRT in marine food webs from 1950 to 2010. As examples, these studies were conducted only by considering species abundance, composition and richness at a globally temporal scale, and the resultant conclusions in time domain were important and valuable, but only provided a coarse view in showing negative influences of the rise of ST on species diversity and populations.

On the other hand, the significant increase of ST is attributed to the gradual accumulation of imperceptible anomalous fluctuations within a season or shorter periods of time, whereas our knowledge on seasonal variation in the composition and population of species communities related to ST is still poor. Scientists have realized the importance of deeply capturing the relationship between marine communities and seasonal changes of ocean warming at a finer scale (considering season or shorter time periods). Some studies have also presented apparent seasonal fluctuations of fish abundance and biodiversity in marine ecosystems by assessing and understanding the composition and seasonal changes of the fish assemblage and doing relevant statistical analyses [14–17]. The results suggested that these seasonal variations could derive from the migration and arrival of species and their own seasonal changes of behavior related to the seasonality of temperature patterns. Although these results provided more specifics for observing linear or non-linear relationships between fish ecosystems and ST, only showing the seasonal variations of fish biodiversity and populations considering sea water

temperature by analyzing the time-series observations of species composition and abundance in a traditional ecological way is still inadequate. Marine fish ecosystems themselves, as a prototype of holistic system, should be treated as an integral whole in a system view so that we can study the resilience, stability and the state evolution of marine ecosystems. This scenario allows us to further understand the feedback-and-response mechanism and how the change of ST affects the fish community in terms of collective dynamics.

## 1.2 Stability, sustainability and management of marine fish ecosystems

Oceans are home to a wondrous array of plants and animals including estimated 20,000 fish species, supporting the "ecological bank" of the planet [18], that supports the global food-web (including humans) and maintaining the stability of global ecosystems from ocean to land and atmosphere via the control of biogeochemical cycles [10]. Although it is impossible to know the exact number of species in the ocean (scientists estimate that more than 80% of oceanic species have yet to be discovered [19]), the number of species which people already knew in the ocean is decreasing. It is urgent and important to keep marine fish ecosystems stable, sustainable and well-managed for the environment and human health. Conventional methods to understand fish community trends are typically analyzing species abundance and/or richness fluctuations in comparison to variations of ST individually. We argue that these approaches, especially when adopting linear models, are far from adequate to deeply understand how fluctuations of ST affects the collective dynamics of fish communities that is deeply rooted into species interactions. The latter are the cause of cascading changes in diversity and abundance. In fish communities, there is a large number of fish species including native and invasive species, preys and predators that interact each other non-linearly. In a dynamical system purview, these species take distinct roles for maintaining the marine ecosystem stable (and sustainable) and always in conjunction with the collective. Therefore, investigating fish ecosystems accounting for ecological-environmental asymmetries and their temporal change is fundamental for understanding feedback between species and environmental determinants, as well as for defining appropriate solutions to improve ecosystem resilience of to ocean warming. Conventional linear models largely fail to capture the non-linearity—arising from time-delayed effects and multiple interactions—to study the response of fish communities to fluctuations of ST. Non-linearity that proves useful to identify the most affected species by temperature change, and the ones mostly responsible for system stability. Even though some studies have investigated the stability of ecosystem using mathematical models [20–23], integrated methodologies that incorporate non-linear network-inferred interaction and predictive models for understanding marine fish ecosystems are still lacking.

Intuitively speaking, biodiversity increases ecosystem stability and sustainability, but climate change or other human-driven perturbations may alter this positive relationship [24]. Therefore, this relationship has been contentious. In fact, the concept of ecosystem stability is complex due to multifaceted understanding and types [25–27]. As for a fish ecosystem, different types of stability describe different features that might lead to different patterns. The stability of fish ecosystems should be explored from the perspectives of species interactions underpinning complex food-webs (when inferred from abundance [28]), the topology (stability) and transitions defining resilience of communities to different types (point-source and diffused) of environmental perturbations [29].

For this purpose, an Optimal Information Flow model (OIF) developed in [28] is used to investigate ecosystem changes over time and temperature and its via metrics of inferred species interactions networks. OIF macroscopically analyzes the evolution of the fish ecosystem over time and ST, and "microscopically" identify critical species who play a significant role in

maintaining functional stability. The dynamical analyses over time, functional stability and salient species assessment is novel with respect to [28]. Analyses are done as a function of time and temperature jointly (e.g. temperature-dependent time variability) and separately (e.g. atemporal seasonal patterns and temperature-independent stability).

## 1.3 Optimal information flow model and multiscale ecosystem analysis

In this study, to overcome limitations of conventional time-domain ecological analyses and explicate how the change of ST fluctuations affect fish communities at multiple scales, we adopt a complex system perspective. Specifically, a fish community in Maizuru Bay managed by the Maizuru Fishery Research Station of Kyoto University [21] is used as a prototype of our proposed information- and network-theoretic models.

Part of the work in modeling fish ecosystems lies in detecting causal interactions between species, that is a fundamental this but challenging task in real-world ecosystems. For this purpose we previously developed the information-theoretic OIF model in which causality is perceived as information flow or cross-predictability of species abundance, that makes causality as a practical computational problem of collective information gain (or uncertainty reduction equivalently) leaving aside all ecological details underlying cross-abundance variability [28]. OIF is based on Transfer Entropy (TE) with time delays that provides an asymmetric approach to measure directed information flow between random variables (species) from time-series data [30, 31], and associates cause-effect relationships with directed information flow considering the extent to which one species improves the prediction of another species' future states in the context of the history of another species [32]. Information flows may relate to different types of interactions, for instance, to the collective behavior of species motion (considering data of relative position and alignment between fish species) [33, 34], or dynamical fluctuations of abundance as in this study. The cause-effect relationship can be interpreted as predictability between species to infer species interactions (also defined as *predictive causality* in [28]) in the fish community and reconstruct networks for describing the marine fish ecosystem, allowing us to study the ecosystem on both temporal and temperature-dependent scales in a systemic way. OIF model is improved with respect to [35] by considering its extension over time to reconstruct dynamical information networks, the varied Markov order of random variables and a refined pattern-oriented criteria to select optimal time delay and threshold based on maximization of mutual information. Therefore, OIF overcomes the limited TE for the directed uncertainty reduction scheme, for the consideration of maximum information (entropy) network [36], and MI-based maximization criteria to define the optimal time delay and interaction threshold to accurately predict systems' patterns (e.g. biodiversity). It is noteworthy to mention that the optimal threshold on interactions is not necessarily within the scale-free or maximum entropy range of inferred collective behavior. In this way, interaction processes are clearly linked to patterns which provide relevance to the inference problem. Additionally, OIF is dependent on the choice of appropriate time delay between variables for TE calculation.

Specifically, to study the effects of ST on the fish ecosystem in a dynamical system purview, the long-term abundance time-series data are categorized into five groups considering five temperature ranges (TR): $\leq 10°C$, $10$–$15°C$, $15$–$20°C$, $20$–$25°C$, $\geq 25°C$. We detect mutual relationships between species and reconstruct networks for each TR to investigate how the fluctuations of ST affect the collective behavior of particular fish species and the whole fish ecosystem by conducting network-based analyses on species interaction networks. For each TR, causal interactions between species are inferred by OIF model (computed as TE considering optimal time delay and threshold). These OIF-inferred species interactions define the connectivity

among species, forming OIF networks for all TRs. Structural and functional patterns are recognized for OIF networks. Network-based species-specific analyses are also conducted by analyzing how much a particular species interacts others, and identifying the most salient links and critical nodes in networks.

Through comparison of functional features of interaction networks (characterizing network topology or structure) as well as of critical species among five TRs, we draw conclusions on the influence of ST fluctuations on the fish ecosystem and its stability. In particular, temperature-dependent but time-independent networks are inferred and analyzed to trace the collective organization of seasonal patterns; this informs about baseline stability, and in particular criticality (as Bak's optimality [37]) in relation to TR network topology. Temporal networks are also implemented to capture how these seasonal patterns change and trigger instability (as critical slowing down a' la Scheffer [38]), diversity, and keystone species when considering long-term ocean warming. The study is novel in providing: (i) connections between abundance and interactions patterns with identification of the most sensitive indicators to predict changes; (ii) non-linear causal attribution of temperature on species and collective stability (including the effect of environment-biota synchronization); (iii) entropic inference of distinct species groups dynamically (competitive and cooperative species); (iv) and interaction-based detection of keystone salient species for ecosystem organization. Thus, because of the novel multiscale analyses (in terms of biota from species to collective, and of the environment from TR to time) linking criticality to critical transitions, the work is providing a potent tool and indicators to understand and track multitrophic stability of marine ecosystems; fish community is an application but any species can be tracked. This is extremely valuable to formulate accurate science-based fishery policy to maintain ecosystems stable and sustainable toward desire states and potentially improve ecosystem resilience to ocean warming.

## 2 Materials and methods

### 2.1 Fish community abundance time-series and seasonal categorization

Long-term time-series data (in total 285 time points) of the fish community were collected by scientists at Maizuru Fisheries Research Station, Kyoto University [21]. They conducted underwater visual census approximately every two weeks along the coast of Maizuru Bay from 1 January 2002 to 2 April 2014 [39]. Data collection for the only species of jellyfish (*A. aurita*) was different: jellyfish were counted from the pier at the Maizuru research station. For jellyfish data, it was count data, but divided by a research area (about 150 m$^2$), so data for this species only have values below a decimal point. Such high-frequency time series enable the detection of potentially causal interactions between species. The fluctuation dynamics of species was later corroborated by eDNA species sampling [40]. Maizuru Bay is a typical semi-enclosed water area with nearly 50 m of the shore and at a water depth of 0–10m, located in the west of Wakasa Bay, Japan. Precipitation is rather high from summer to winter that is the rainy season in this area. Sea-surface temperature ranged from 5.2 to 31.8˚C, sea bottom temperature from 8.5 to 29.6˚C, which were measured near the surface and at the depth of 10 m underwater during the diving, respectively [41].

In this dataset, only 14 dominant fish species whose total observation counts were higher than 1000 (considering non-normalized abundance [21]), and 1 jellyfish species were included because rare species were not observed during most of census period. Rare species would bring large numbers of zeros in the time series which may lead to difficulty in analyzing data. As proven in other similar studies, ignoring rare species (in average abundance and without extreme fluctuations) does not significantly change fish ecosystem results since abundance dynamics of rare species contained within other species is typically very small; thus, the

inclusion (or exclusion) of rare species does not alter the inferred ecosystem interaction topology beyond just adding poorly interacting species as found in [21, 28]. Jellyfish, a non-fish species, was involved in the dataset since it was thought to have prominent influences on the dynamics of the fish community due to its large abundance. Note that time-series data were normalized to unit mean and variance before analysis [21, 42] (i.e., technically normalized to zero mean and unit variance); this is done to ensure all species have the same level of magnitude for comparison and to avoid constructing a distorted state space.

In this study, we aim to study the effects of ST change on ecosystem dynamics and stability considering in particular how slow ST increase affects seasonal species organization and how that implicates long-term changes. This is done because ecosystems have memory and ST gradual changes accumulate over time leading to irreversible changes in the short-term dynamics (manifesting gradually or suddenly [38]), potentially. Initially, the normalized time-series abundance were categorized into five subsets considering five mean temperature range (the average of surface and bottom water temperature was considered), i.e. $\leq 10\,°C$, $10–15\,°C$, $15–20\,°C$, $20–25\,°C$, $\geq 25\,°C$, resulting in five shortened time series with 16, 93, 58, 62, 56 time points. This allowed us to disentangle how the fish community fluctuates seasonally with the change of ST by separately analyzing time series of these five subsets. This is done independently of time considering the whole period to assess potential seasonal stability via probability distribution or network topology equivalently [43]. The data categorization picks the abundance observations at time points when the mean ST is within specific TRs and rearranges the selected abundance observations by preserving the time sequence, leading to new time series corresponding to TRs. The subset of time series after categorization in consideration of five TRs are still regarded as sequential time series, called TR-dependent time series here, even though the between-TR abundance interdependence is not considered as in the continuous temperature-dependent analysis (see Section 2.5).

## 2.2 Probabilistic portrayal of the fish community: Power-laws and transitions to exponential

We estimate the probability distribution function of normalized fish abundance and of inferred interactions between species (both are indicated as $y$ as a generic random variable) using power-law or exponential distribution functions [44, 45]. Theoretically, the power-law function that has been forced a-priori (leaving the exponential distribution as the alternative function in case of non-fitting via Maximum Likelihood criteria on Kullback-Leibler divergence) has two types: discrete and continuous, of which the continuous form [46] is given by:

$$p(y) = \frac{\gamma - 1}{y_{min}} \left( \frac{y}{y_{min}} \right)^{-\gamma}, \tag{2.1}$$

where $y_{min} \geq 0$ is an estimated lower bound or cutoff for which the power-law starts to hold. $\gamma$ is the power-law scaling exponent underlying the criticality of the studied variable [37]. $\gamma = \epsilon$ or $\Phi$ if $Y = X$ (abundance) or $TE$ (interactions). The power-law exponent indicates how mean, variance and higher statistical moments are defined (finite or infinite depending on $\gamma$ [47]). Specifically: for $\gamma \leq 2$ the regime is critical with mean and all moments that are infinite (accurately, this is the only heavy-tail distribution that almost always has finite-size effects); for $2 < \gamma \leq 3$ the regime is supercritical with finite mean and all higher moments as infinite; and, for $\gamma \geq 3$ mean and all moments are finite. The latter regime is the last power-law regime before the exponential one. Strictly, power-laws should be defined for two orders of magnitude at least but we relaxed this condition due to data sparsity in this case study.

Distributions of species variables (abundance and interactions) are visualized by computing discrete exceedance probability distribution (epdf) that is defined as $P(Y \geq y) = 1 - P(Y < y)$, where $P(Y < y)$ is the cumulative distribution function (cdf) [46] derived from probability distribution function $p(y)$ (pdf), and by plotting the epdf on a log-log scale to emphasize the non-linear power-law dynamics useful to estimate the scaling exponent. Exponents were estimated using the approach and package of [48]. We also introduce the cutoff $Y_{break}$ to the random variable whose probability distribution explicitly presents multiple regimes. In our study these probability distribution regimes are power-law and occur with different exponents that are separately estimated. This double Pareto distribution was highlighted in different ecosystems and analytically formulated by [47]:

$$P(Y \geq y) \sim \begin{cases} c_1 \, y^{-\gamma_1 + 1} F\left(\dfrac{y}{y_{min}}\right) & \text{for } y_{min} < y < Y_{break} \\ \\ c_2 \, y^{-\gamma_2 + 1} F\left(\dfrac{y}{y_{break}}\right) & \text{for } y \geq Y_{break} \end{cases}, \qquad (2.2)$$

where $Y_{break}$ is the break point isolating two power-law distribution regimes, $c_1$ and $c_2$ are two coefficients, $F(y)$ is a homogeneity function [47], $\gamma_1$ and $\gamma_2$ are power-law exponents of the two regimes. This "broken power-law" function (or double-Pareto distribution function) is a piece-wise function consisting of two or more components with different power-law scaling exponents separated by break points [47, 49, 50]. Specifically in this study, a break point was set to divide the epdf of species interactions for the 15–20°C TR. The break point $Y_{break}$ estimated from the model was much higher than the median value of species interactions for the first power-law regime. If a power-law distribution of abundance or interaction was not feasible, then an exponential distribution was fitted to the data, i.e. $P(Y \geq y) \sim e^{-\lambda y}$.

## 2.3 Species diversity and abundance characterization

To characterize the fish community macroecologically, $\alpha$-diversity is introduced to describe the fluctuations of species diversity over time and temperature. $\alpha$-diversity refers to the biodiversity within a particular scale (time period, area or TR) and is computed as the number of species in that scale. Given a set of unique species $\mathbf{S} = \{S_1, S_2, \ldots, S_n\}$ whose normalized abundances $\mathbf{X} = \{x_1, x_2, \ldots, x_n\}$ change over time, time-dependent $\alpha$-diversity $\alpha(t)$ is defined as in [35]:

$$\alpha(t) = \sum_{i=1}^{n} x_i(t)^0, \qquad (2.3)$$

where $x_i(t)$ is the normalized abundance of species $i$ at time point $t$. For each time point, sea surface temperature and bottom temperature were recorded. We average these temperatures as mean ST and reorganized $\alpha$-diversity dependent on the mean temperature yielding temperature-dependent $\alpha$-diversity $\alpha(T)$.

Additionally, fish species are distinguished into fish stocks (*FS*) [51, 52], native and invasive species groups as shown in S1 Table. This categorization was based on the information retrieved from *Fishbase* (http://www.fishbase.org/). We separately calculate the total abundance of all species, defined as Ecological Productivity (*EP*) that is proportional to the total biomass [53–55], and of these species groups, FS, native and invasive dependent on time and

temperature as:

$$A(T) = \sum_{i=1}^{n'} x_i(T), \qquad (2.4)$$

where $T$ can be time or temperature, $x_i(T)$ is the normalized abundance of species $i$ at time or temperature $T$, $n'$ is the number of species in a particular subgroup (for all species group EP, $n'$ is $n$ that is 15s).

## 2.4 Information-theoretic patterns and network inference

**2.4.1 Abundance uncertainty.** Entropic diversity indices are capable of providing ecological information on the patterns of assembly including rarity, commonness and fluctuation of abundance of species in ecosystems in a probabilistic way vs. taxonomic diversity. In this study, we calculate Shannon entropy for each species in the fish community based on pdf estimation of species abundance. Shannon entropy of species $i$ is defined as $H(X_i) = -\sum_{m=1}^{v} p(x_i(m)) log_2 p(x_i(m))$, where $p(x_i(m))$ is the probability of an event $m$ of unique normalized abundance of species $i$. $v$ is the number of all unique events of abundance. Note that event is a concept in probability theory, here we considered a unique number of species abundance observed in the time-series data as an event. The species-specific Shannon entropy implies the uncertainty of species observations (samples) over time, and is used as one of information-theoretic variables to define the structure of inferred networks (the size of nodes).

**2.4.2 Optimal information flow networks as collective abundance uncertainty reduction.** To investigate the collective behavior of the fish community, we estimate predictive causal interactions between fish species by using the Optimal Information Flow (OIF) model [28] developed by our group. Collective behavior is here analyzed by extracting networks dependent on (i) continuous time; and (ii) continuous temperature and discrete TR groups (without between TR-abundance interdependencies). The former and the latter are for mapping long-term and seasonal ecosystem organization, and time dynamics conditional to temperature is also performed for identifying clearly which season and species are affected the most by ST increase.

We quantify information processing of environmental change in species assimilated in community abundance (due to individuals affected) that also contains information processing of baseline species interactions. The role of the environment is likely multiplicative than additive; yet, non-linear assessment of collective interaction change is the most suitable way for extracting environmental change effects. Transfer Entropy (TE) or information transfer, a quantity in information theory coined in [30], is extensively used to measure information flow between two variables and is considered as one of the most diffused non-linear methods to quantify causal interactions. TE is commonly used as a powerful analytics to estimate mutual interactions in non-linear ecosystems due to its explicit consideration of probabilistic asymmetry between variables leading to their mutual predictability without making any further assumptions on underlying ecosystem processes [28, 56–59]. TE in an energetic sense is actually energy consumption per time unit (or metabolic flux) between species [60]. Yet, ecologically TE is a proxy of the energy involved in abundance-related interactions underpinning food-webs.

TE is mathematically defined as the amount of information that a source variable provides about the next state of a target variable in the context of the past of the target [30]. It provides a prediction-oriented and probabilistic-based tool in detecting directed and dynamical causal interactions without the need of any prior knowledge on mechanisms or assuming particular

functional forms to describe mutual interactions among elements in a dynamical system. As a result, we depict "causal interaction" as predictability of collective dynamics (or directed uncertainty reduction, equivalently) that is easier to interpret and mathematically formalize [28]. The calculation of TE between two variables builds on conditional and joint probabilities considering Markov order and historical values of these variables (i.e., the self/independent and co-dependent predictability). TE is formulated as:

$$TE_{X_i \to X_j}^{(q,s,u)} = \sum_{x,y} p(x_j(t+1), x_j^{(s)}(t), x_i^{(q)}(t+1-u)) log \frac{p(x_j(t+1)|x_j^{(s)}(t), x_i^{(q)}(t+1-u))}{p(x_j(t+1)|x_j^{(s)}(t))} \ , \qquad (2.5)$$

where $X_i$ and $X_j$ stand for two random variables, $q$ and $s$ denote the Markov order of variables $X_i$ and $X_j$, $x_i(t)$ and $x_j(t)$ are time-series observations, and $u$ is the free varied source-target time delay that yields time lagged interactions. In this study, $X_i$ and $X_j$ are normalized abundances of species $i$ and $j$ in the fish community. TE assumes that all analyzed variables obey memory-less stochastic Markov chain process [61]. It indicates that next states of variables are only dependent on the current state, while not determined by states in the past. Thus, $q$ and $s$ are fixed as 1 under this assumption. TE calculation is sensitive to data features including probability distribution estimation, extreme values and zeros. In this study, we apply Kernel and Gaussian estimators (parameter-free) to test the dataset of the fish community [62] and compare the results with the ones from CCM model that is another well-documented algorithm to measure causal interaction between variables [63], and the Pearson correlation coefficient.

After calculating all TE pairs the subsequent steps of OIF [28] were implemented, i.e. detection of direct links and simplification of the network without altering collective predictability (or systemic uncertainty reduction equivalently).

Species interaction networks are reconstructed from TE matrices to visualize directed interdependencies between species. As the networks are hard to distinguish due to large numbers of connections, we first select an appropriate time delay $u$ that minimizes the statistical distance between species defined as Eq 2.6 [64], equivalently maximizes the mutual information between species, to reduce redundant information.

$$d(X_i, X_j) = e^{-I(X_i; X_j)}, \qquad (2.6)$$

where $I(X_i; X_j)$ is mutual information (MI) between species $i$ and $j$. MI is given in information theory as:

$$I(X_i; X_j) = \sum_{x_j} \sum_{x_i} p(x_i(t), x_j(t)) log \frac{p(x_i(t), x_j(t))}{p(x_i(t))p(x_j(t))}, \qquad (2.7)$$

where $p(x_i(t))$ and $p(x_j(t))$ are marginal distributions of species $i$ and $j$ and $p(x_i(t), x_j(t))$ is joint distribution of these two species. In this study, we specify the time delay $u$ ranged from 0 to 10. According to Eq 2.7, time delay $u$ is selected as an integer in the range that maximizes the mutual information. It implies that $u$ is determined by considering predictability rather than "true" causality which is strenuous to concretely estimate. The maximum TE (after Eq 2.6 selecting the most likely time delay) was also used in this study to measure the inuence of temperature on species and temperature on ecosystem stability.

Afterwards, an appropriate value is selected as a TE threshold considering the maximization of prediction of $\alpha$-diversity via effective $\alpha$-diversity (Eq 2.10 in Section 2.5). This is the second step to do the redundancy reduction and simplify the structure of inferred networks by removing weak interactions (links) in networks. Note that this criteria is arguably one of the most important in reproducing macroecological patterns of species richness variability; however,

other criteria are valid such as thresholding TE based on the "80–20" Pareto principle to identify interaction cores in communities.

The threshold step is analytically formulated as:

$$f(X_i \rightarrow X_j) = \begin{cases} TE_{X_i \rightarrow X_j} & \text{for } TE_{X_i \rightarrow X_j} \geq TE_{thre} \\ 0 & \text{for } TE_{X_i \rightarrow X_j} < TE_{thre} \end{cases}, \tag{2.8}$$

where $f(X_i \rightarrow X_j)$ is the quantification of species interactions from species $i$ to $j$, $TE_{thre}$ is the threshold chosen to filter TE values. In this study, $TE_{thre}$ can be a fixed value 0.01 or top 20% TE value [65]. These steps of MI-based time delay identification, TE estimation model selection considering data probabilistic dynamics, coupled maximization of uncertainty reduction and removal of indirect links and optimal TE threshold selection form the proposed OIF model [28].

## 2.5 Continuous time- and temperature-dependent dynamical network analysis

Beyond the TR-constrained "seasonal" analysis (independent of time), OIF [28] was used to extract dynamical networks considering time and temperature as continuous dependent variables to analyze the ecosystem in long-term and for smaller steps along the temperature gradient. By necessity this is done by truncating the whole time series in time and temperature units among which networks are reconstructed. Discrete TR networks (for the groups in Section 2.1) are essentially extractable from the continuous analysis.

The total number of time-series units (number of networks and community features) on which dynamical network inference is performed by OIF is $G = \lfloor \frac{L-l}{\Delta l} \rfloor + 1$, where $\lfloor \bullet \rfloor$ rounds $G$ to the smaller integer. When reconstructing time-dependent dynamical networks, $L$ is the total number of samples ($L = 285$, i.e. the whole time period), $l$ is the length of each time-series unit (time unit hereafter), $\Delta l$ is the chosen inter-observation gap (or time step). In this study, each time period is set as one year (the length of time-series unit $l$ is 24 time points that guarantees to have enough points for reliable TE inference), time step $\Delta l$ is 2 time points that correspond to one month, and then the whole time series is truncated into 131 time-series units in total where networks are inferred. The overlap (and yet dependence of species) of data/networks is nine months vs. 2 weeks of [21] and we believe this is a more proper choice considering that species do not alter dynamics in a matter of weeks and more importantly independently of previous historical trends (earlier than two weeks from the present). In this regard, our approach considers long-term dynamics ($l = 1$ year data) for estimating short-term features at a scale $\Delta l = 1$ month. This is also the scale at which stability and effective diversity were calculated. The non-linearity between species is considered via time-delayed co-predictability of pdfs (i.e. TE); yet, more deeply than [21] that linearizes the non-linear community variability by considering ratios of delayed species abundance every two weeks. The latter is the basis of CCM [21], typically estimating local feature from local dynamics only (and without pdfs), unless larger time delays are selected but those cannot be too large without compromising computational efficiency.

When reconstructing temperature-dependent networks considering temperature as a continuous variable vs. discrete TR analysis (Section 2.1), $L$ is the maximum temperature rounded to the largest integer (in this dataset the maximum mean temperature is 30.7°C, thus $L$ is 31), $l$ is the upper limit of the first temperature range (for convenience we picked the range $[5 - 10]$°C and then $l = 10$°C) and $\Delta l$ (set to 1°C) is the temperature step establishing the moving window (or the difference between networks over temperature, equivalently). Therefore,

the whole time series is truncated into 22 temperature units (calculated as $G = \lfloor \frac{L-l}{\Delta l} \rfloor + 1$) corresponding to different temperature ranges whose difference is 1˚C (and overlap is 4˚C) for providing small-scale estimates by considering continuous smooth transitions along the temperature scale. Note that the temperature unit size (or range), over which temperature-dependent networks are estimated, is set to $l^* = 5$˚C that is granular enough to describe ecosystem variations. This is in analogy to the predictability overlap concept used for the temporal network inference: species interactions do not change abruptly for every Δ-temperature and yet non-linear superposition of species along the temperature-scale should be considered. The length of time series for different temperature units (representing somehow temperature niches) is different due to the fact the number of species in each unit can vary; note that the temporal sequence of abundance for different temperature units is preserved. This is different than temperature analysis where the number of samples every time units is always the same because of sampling design; yet the time unit size corresponds to the lower limit $l$. By selecting temperature units equivalent to the defined TRs in Section 2.1 it is possible to derive the discrete seasonal networks. Lastly we also performed TR-constrained dynamical analysis over time by combining time-dependent data conditional to the discrete TR groups defined in Section 2.1.

After time and temperature-dependent reorganization (i.e. picking abundance data within all temperature ranges), OIF is used to infer species interaction (TE) for each time-series unit (time or temperature unit), resulting in temporal and temperature-dependent interaction matrices $W_{i,j}(g)$ and dynamical networks. Here, $g$ is the time point of each time-series unit for temporal networks, while the lower limit of the temperature range for temperature-dependent networks. We analyze dynamical networks by computing total interaction (TI), local dominant eigenvalues and estimated effective $\alpha$-diversity of each dynamical network. TI is defined as the sum of all TE asymmetrical interactions in the interaction matrix $W_{i,j}(g)$, i.e.:

$$TI(g) = \sum_{i,j} W_{i,j}(g). \tag{2.9}$$

The estimated effective $\alpha$-diversity $\alpha_e(g)$ is defined as the number of nodes (species) involved in a dynamical network [28] that is formulated as:

$$\alpha_e(g) = \sum_{i=1}^{n} h_i(g), \tag{2.10}$$

where

$$h_i(g) = \begin{cases} 0, & \text{for } \sum_{j=1}^{n}(|W_{i,j}(g)| + |W_{j,i}(g)|) = 0 \\ 1, & \text{for } \sum_{j=1}^{n}(|W_{i,j}(g)| + |W_{j,i}(g)|) \neq 0 \end{cases}. \tag{2.11}$$

Therefore, $\alpha_e(g)$ denotes the total number of nodes whose functional degrees (considering asymmetrical interactions [35]) are not zero in a dynamical network. We use absolute values of $W$ in Eq 2.11 for generality, in case interactions are generated by a model that identifies positive and negative interactions such as in CCM [21]; note that in OIF any negative TE value does not have physical sense (or is mathematically plausible, unless a negative sign is assigned a posteriori to opposite interaction directions between $i$ and $j$) [28] and should be considered as a numerical artifact to remove.

**2.5.1 Network-based spectral ecosystem stability and species salience.** The interaction matrix, defining the connectivity of dynamical ecosystems, is widely used to analyze ecosystem stability considering the whole dynamics and complexity [21, 22, 66–68]. It also defines the

topological transitions of ecosystems if tracking their interaction topology, allowing us to analyze them in a systemic view. Furthermore, interaction matrices extracted from time-series data of species abundance capture the functional information of complex ecosystems beyond the dynamics of species abundance and richness themselves. Therefore, we study ecosystem stability by analyzing eigenvalues of the TE matrix and its distribution in the complex plane. In this study, species interactions are inferred by OIF [28]. Let the matrix $W$ be the TE matrix, if there is a vector $v \in \Re^n \neq 0$ that satisfies:

$$Wv = \xi v, \tag{2.12}$$

then scalar $\xi$ is called the eigenvalue of $W$ with corresponding right eigenvector $v$. The real part of eigenvalues determines whether and how fast the network returns to equilibrium from a perturbation [68], yet it is a signature of ecosystem time-point resilience; more negative values indicate that the community returns to stability more quickly. The imaginary parts of eigenvalues predict the extent of oscillations in species abundance during the return to equilibrium; larger imaginary components imply more frequent oscillations.

It should be noted that our approach is different than the spectral approach in [21] in several ways. In particular we chose $W = TE$ as the matrix reflecting "interactions strength" that is proportional to what the theoretical Jacobian should be for canonical eigenvalue. In [21] the Jacobian is construed on local interspecific and intraspecific interaction strengths that are essentially the derivatives of abundance of two or the same species at different time steps (where the maximum delay is taken as two weeks). We argue that this very local assessment in [21] determined the very regular and periodical trend in dominant eigenvalue. In our context interaction strength is TE so conceptually the approach is equivalent with the difference of the eigenvalue rescaling due to the lack of consideration of ΔTE. Additionally in our approach the overlap between interaction networks is nine months by construction (Section 2.5); yet, a long-range interdependence is considered when assessing adjacent *TE*s on which more global eigenvalues are determined.

Dynamical Stability (DS) (a' la [38]) is calculated as the absolute value of real part of dominant eigenvalue, that is the local Lyapunov stability [21, 22], as:

$$DS(g) = |Re(\xi_{max}(g))|, \tag{2.13}$$

where $\xi_{max}(g)$ is the dominant eigenvalue of an interaction matrix. Note that on the contrary of [21] we do not remove seasonality from the normalized abundance data and the dominant eigenvalue is assessed for all dynamical networks whose data overlap except for a two-week gap of the moving window used in network inference (see Sections 2.1 and 2.5). Yet, there is no complete independence between time units, that are two-week units, as in [21]. A dynamic stability value of less than 1 (in discrete-time linear dynamical systems) or less than 0 (for continuous-time nonlinear dynamical systems as it is more general for ecosystems) indicates that the community tends to recover faster from perturbations; higher values than 1 or 0 imply higher instability and values of 1 or 0 identify critical transitions. However, absolute values are less meaningful than dominant eigenvalue trends and fluctuations because those are highly subjected to data in input (e.g. normalized or not according to different normalization schemes, such as initial or maximum value or zero mean and unit variance, and how the interaction matrix $W$ is built). The eigenvector corresponding to the dominant eigenvalue become slower as the dominant eigenvalue of the interaction matrix (ideally the Jacobian) approaches zero (for continuous non-linear dynamical systems) [67]. In many models that represent the critical transition as a transition from one local stable equilibrium to another, the transition is preceded by decreasing asymptotic stability of the equilibrium from which the transition

occurs. This mathematical phenomenon is referred to as critical slowing down (CSD) and we evaluated that for the whole time period in order to assess global stability (or instability loss equivalently) vs local stability. Locally the equilibrium is stable if the real part of all of the eigenvalues are negative and unstable if any real parts are positive. CSD occurs as one of the eigenvalues approaches zero and becomes less negative and thus the return to the equilibrium becomes slower. In a broader sense, CSD has been associated in the past to changes in information propagation length [68, 69] (based on TE) and Ricci curvature of networks underpinning systemic risks [70]: metrics that can be used analogously to the dominant eigenvalue.

To evaluate ecological importance of species based on OIF networks, we innovatively identify critical species in the fish community (indicated as nodes in OIF networks) using information-theoretic indices and link salience. The information-theoretic index is defined as total outgoing transfer entropy (OTE, computed as $OTE(i) = \sum_j TE_{i \to j}$) [35] that measures how much one species affects others totally. $OTE(i)$ is the total information transition from species $i$ transmits to all other species. It can be therefore interpreted as how much one species helps to predict others in terms of predictability. $OTE$ index is able to measure species influences with directions thanks to the asymmetric property of TE.

We also employ link salience that was introduced in [69] to measure the importance of links in OIF networks. Link salience approach is based on the concept of effective distance $dist(i, j)$ (computed as $dist(i, j) = 1/W_{ij}$). It is intuitively assumed that strongly (weakly) interacting nodes are close to (distant from) each other. In our heterogeneous networks with real-valued weights, the algorithm of the shortest-path tree ($SPT$) that identifies the most efficient routes from a reference node $r$ to the rest of the network is implemented for all nodes ($SPT(r)$). Then $SPT(r)$ is represented by a $n \times n$ matrix with element $spt_{ij}(r) = 1$ if the link $(i, j)$ is involved in the collection of shortest paths, and $spt_{ij}(r) = 0$ if it is not. In conclusion, the link salience of OIF network is defined as:

$$SAL = \frac{1}{n} \sum_{r=1}^{n} SPT(r), \tag{2.14}$$

Therefore, $SAL$ is a linear superposition of all $SPTs$. If the element $sal(i, j) = 0$, link $(i, j)$ has no role in networks; if $sal(i, j) = 1$, link $(i, j)$ is important for all reference nodes; and if $sal(i, j) = 1/2$, link $(i, j)$ is important for only half of reference nodes [69]. Node (species) importance is quantified by counting the frequency that one node exists in the most salient links as the reference terminal node (species).

## 3 Results

All models, scripts for visualizations, data and results are at https://github.com/HokudaiNexusLab/FishCommunity.

### 3.1 Time and temperature-dependent abundance analyses

A simple analysis for original data of fish species abundance starts by looking into temporal trajectories and seasonal fluctuations of sea water temperature (Fig 1A), taxonomic $\alpha$-diversity (Fig 1B) and abundance (Fig 2). It is evident that sea surface and bottom temperature, $\alpha$-diversity of the fish community fluctuated over time synchronously and seasonally. Approximately since 2007, seasonal fluctuations of sea surface and bottom temperature were slightly getting higher except for 2009 with the least fluctuation (Fig 1A), while the global trend of $\alpha$-diversity was decreasing (Fig 1B). This result implies that the change of biodiversity loss in the fish community may relate to the increasing fluctuations of ST. Total abundances of EP, FS, native and invasive groups over time (Fig 2A and 2B) show a slight increase with fluctuations since 2007,

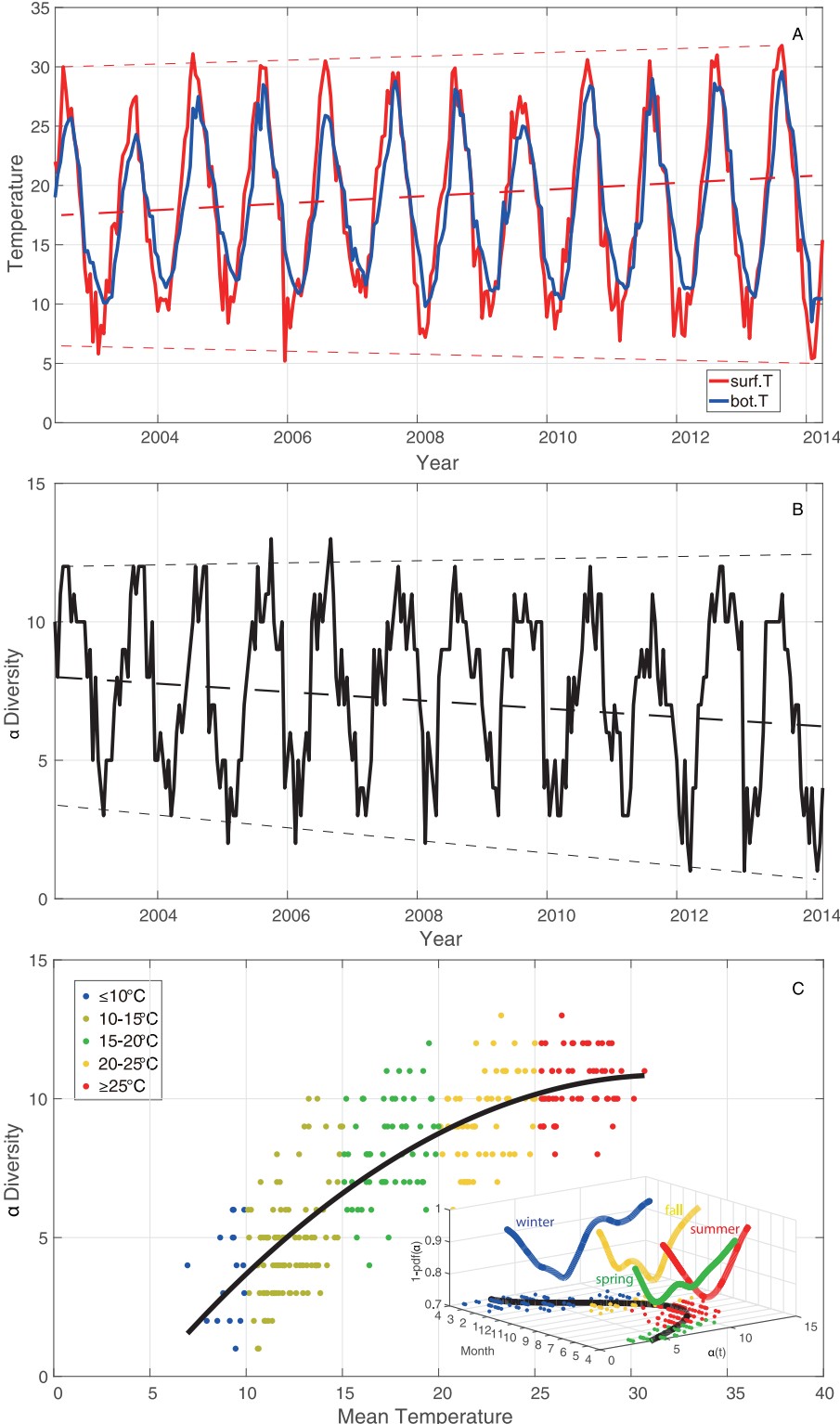

**Fig 1. Seasonal fluctuations of ST and α-diversity and relationship with mean temperature. A**: The sea surface temperature (red line) and bottom temperature (blue line) from June 2002 to April 2014. **B**: Taxonomic α-diversity over time. **C**: α-diversity over mean temperature (the average of sea surface and bottom temperature). Blue points, light green points, green points, yellow points and red points represent values of α-diversity corresponding to different TRs: ≤10˚C, 10–15˚C, 15–20˚C, 20–25˚C, ≥25˚C, respectively. The black curve in plot C is a second degree

polynomial fitting for $\alpha$-diversity over temperature; that establishes an empirical Pareto optimal frontier for the climate niche of fish diversity exerted by temperature. The inset is showing the potential ecosystem landscape across the whole period where the entropy is proportional to $1 - pdf(\alpha)$; transition seasons shows the highest entropy followed by summer.

that is opposite to the global trend of species diversity, with the exception of a spike in 2009 and a downward spiral in 2011. This finding implies that increasing ST makes some species more abundant in global view, even though it negatively affects species diversity, and the latter is very sensitive to the change of ST fluctuations via species interaction alteration.

These results can be more explicitly confirmed in Fig 2 where EP, FS, total abundance of native and invasive species and species 1 (*Aurelia aurita*) are observed to increase exponentially over the seasonal increase in ST [70, 71]. The abundance of species 2 (*Engraulis japonicus*) that is a cooperative species with large geographic range (S15 Fig vs. S16 and S17 Figs for competitive species), as an exception, decreases with the rise of ST both seasonally and over time. It should be carefully noted that patterns over temperature (Fig 2B and 2D) characterize the dynamics of species only across seasons and not over time; rather, changes in the exponential growth parameter over time are meaningful for tracking temporal changes. Therefore, Fig 2B and 2D characterize the baseline exponential response ($X = X_{min}\exp^{kT}$ where $k$ is the growth factor in the legend and $X_{m}$ *in* is the abundance for the lowest temperature) of abundance across seasons. Discordance between these patterns and patterns over time for increasing temperature (e.g. for native species and fish stock) are warning of changes in species related to

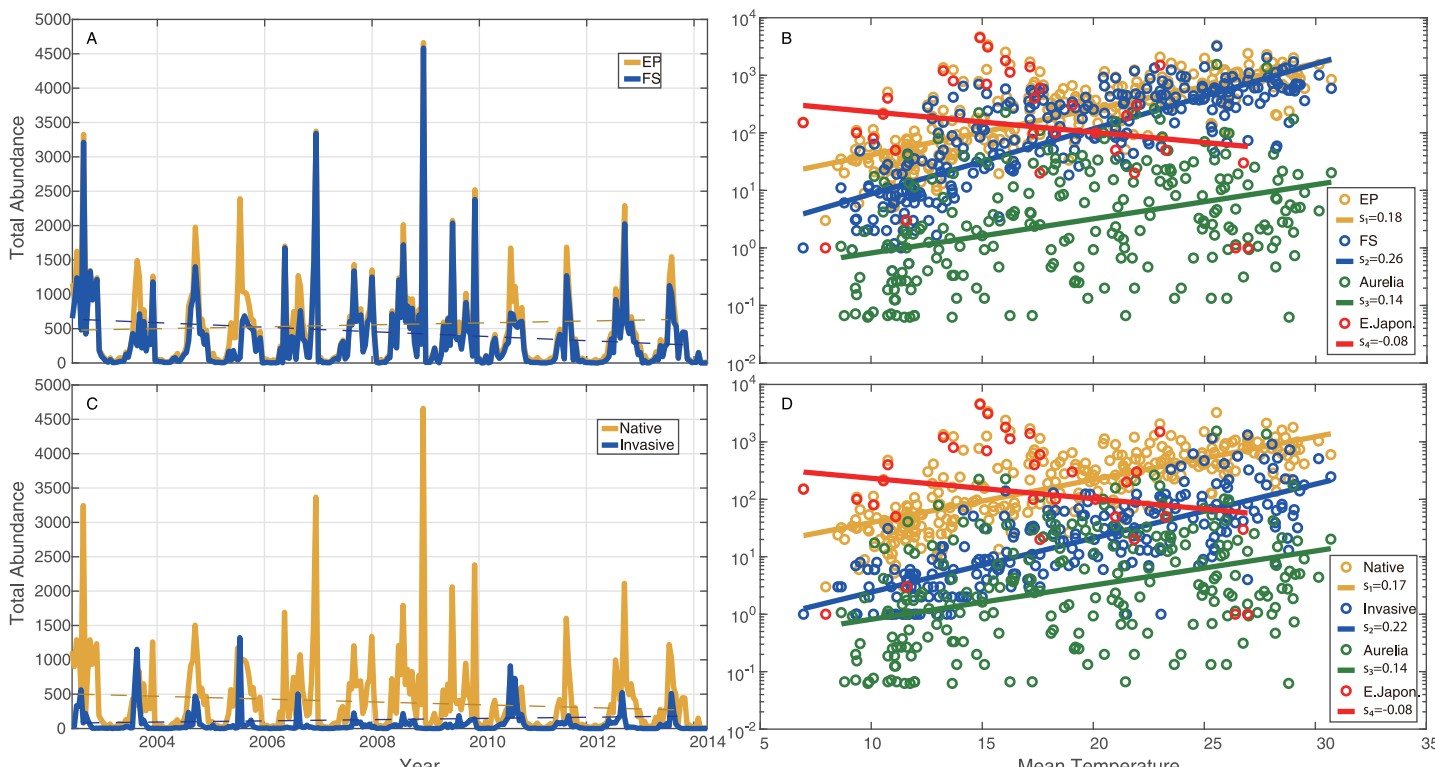

**Fig 2. Total species abundance of EP, FS, native and invasive species over time and mean temperature.** Total species abundance of EP (yellow) and FS (blue) species over time and temperature (**A** and **B**). Total species abundance of native (yellow) and invasive (blue) species over time and temperature (**C** and **D**). Abundance of species 1 (green) and 2 (red) over mean temperature (**C** and **D**).

**Table 1. Observed changes in ecosystem community and population metrics.** Metrics are evaluated considering 2002 and 2014 observed values. Negative Δ % are for decrease in values over time. Δ/year % are the yearly changes over 12 years. Ecosystem stability is based on the real part of the dominant eigenvalue of the whole community (S12A Fig).

| Ecosystem Metric | Δ % | Δ/year % |
|---|---|---|
| Temperature | 20.00 | 1.67 |
| Ecosystem Productivity | -64.30 | -5.36 |
| Fish Stock | 22.45 | 1.87 |
| Native Sp. | -50.00 | -4.17 |
| Invasive Sp. | 150.00 | 12.50 |
| *E. japonicus* (J. anchovy) | -64.00 | -5.33 |
| *T. japonicus* (H. mackerel) | 12.50 | 1.04 |
| *H. tenuispinnis* (C. wrasse) | 116.70 | 9.72 |
| $\alpha$-diversity | -25.00 | -2.08 |
| Ecosystem stability | -24.00 | -2.00 |
| Total Interactions | -14.52 | -1.21 |

other factors, such as changes in species interactions. Table 1 shows the observed ecosystem response to temperature over time considering abundance and other metrics discussed hereafter.

We also reorganize $\alpha$-diversity considering ST, yielding $\alpha$-diversity against temperature shown in Fig 1C. It shows that $\alpha$-diversity grows with the increasing temperature (over seasons), while the rate of the increase of $\alpha$-diversity ($\alpha'(C)$) gradually declines (see the black line fitting the $\alpha(T)$ points in the plot). The decreasing $\alpha'(T)$ implies that fish diversity does not continuously grow with the increase of temperature. Within lower TRs, the fish community is more sensitive to the change of temperature relative to higher TRs.

Time series of normalized abundance are categorized into five groups considering five TRs: $\leq 10°C$, $10–15°C$, $15–20°C$, $20–25°C$, $\geq 25°C$. Epdf for these five TRs are visualized in Fig 3A. Epdf plots show that the distribution of abundance, and hence abundance uctuations, becomes more scale-free with the increase of temperature; note that this is a seasonal analysis that does not reveal any temporal change. Lower power-law scaling exponent for higher TRs means fatter tail of power-law distributions (with slower decay than exponential distribution, yet less resilient Scheffer sensu). This indicates that the distribution of species abundance is less even since some species become exceedingly more uctuating in abundance (with "black swan" extremes) that may or may not be persistent over time (see S1 and S2 Figs show the epdf of abundance for each species individually and competitive/cooperative groups). Power law implies less evenness (the similarity in abundance among species is lower despite common species are more even) but higher long-range organization and this is a beneficial properties of biodiversity, also in an energetic sense considering the lowest entropy of scale-free organization [35]. This is particularly evident for cooperative species (in terms of distribution rather than extreme values) as also show in [28] and Fig 8. These results suggest that for higher temperature, fish species present wider distribution in abundance compared to lower TRs. Fig 3B shows the abundance-based Taylor's law, i.e. standard deviation against mean abundance for each species corresponding to each TR. The regression of log standard deviation vs. log mean abundance for all TRs gives scaling laws with slopes less than 1. Positive slopes address that the total variability of species abundance increases with the rise of mean species abundance. Slopes less than 1 suggest that the per capita variability of species abundance for all TRs decreases with the increasing mean species abundance [72]. Therein, slopes of scaling law for lower TRs ($\leq 10°C$, $10–15°C$, $15–20°C$) slightly increase with the increasing ST, and are obviously lower

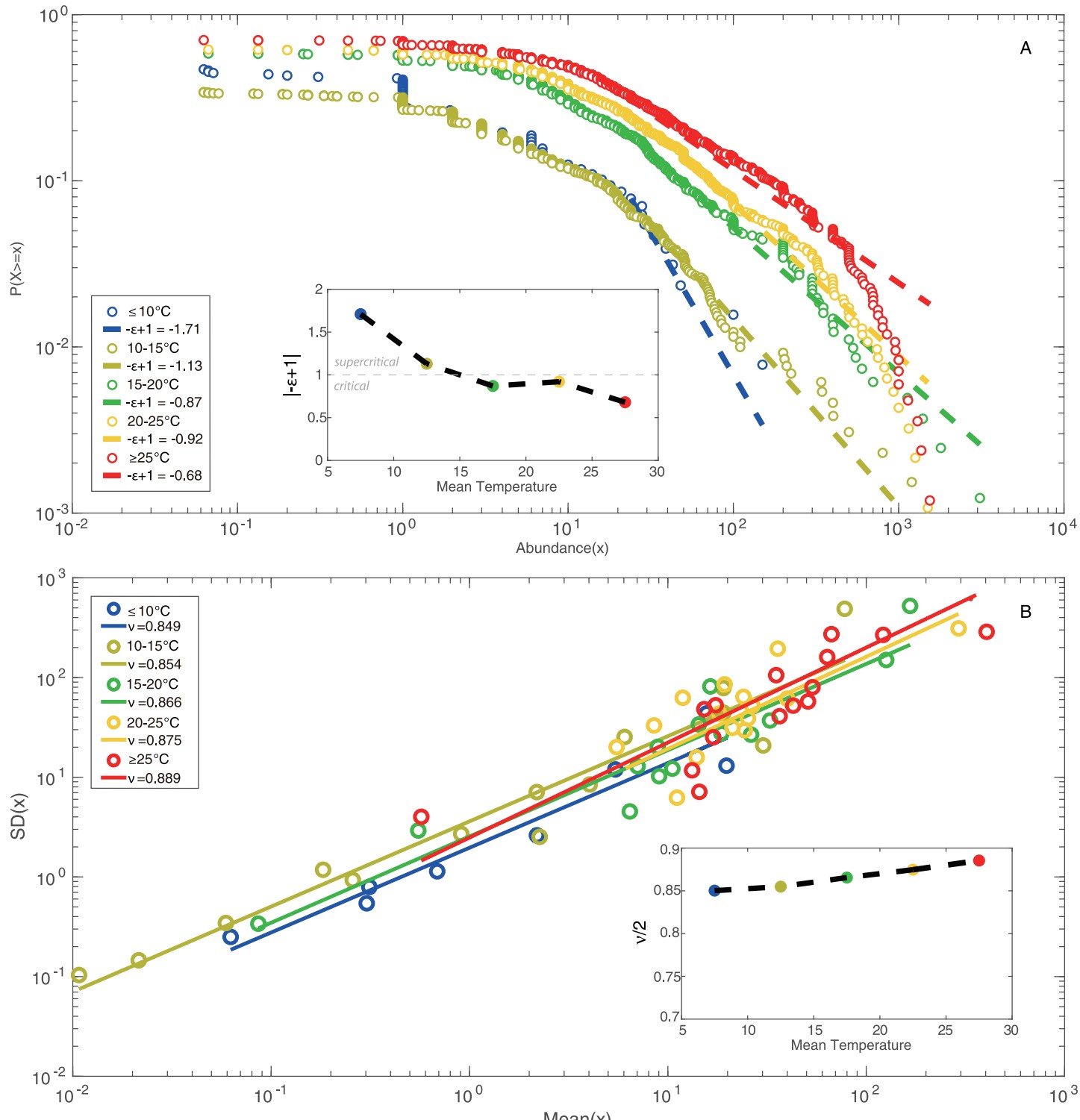

**Fig 3. Zipf's and Taylor's law of species abundance for temperature groups. A**: Exceedance probability distribution functions (epdf) are fitted by power-law distributions. $|-\epsilon + 1|$ is the scaling exponent of the distribution. **B**: Taylor's law as scaling law between standard deviation and mean species abundance for five TRs; $v/2$ is the scaling exponent of Taylor's law that is typically construed by using the variance [72], $\langle x^2 \rangle \sim \langle x \rangle^v$; smaller $v$ means that fluctuations in abundance are more even and species are more regularly distributed; vice versa, species are more power-law distributed (as supported by Zip's law) driven by stronger environmental effects [85] (in this case ST) determining portfolio effects with potential stabilizing effects [86] (this however neglects interaction topology). All exponents for five TRs are interpolated by black dashed lines in insets to emphasize abundance patterns transitions across TRs/seasons; transitions show a gradual second-order phase transition with higher Pareto-distributed fluctuations for higher temperature.

than those for higher TRs (20–25˚C, $\geq$25˚C). It indicates that the variability of species abundance is on average higher for higher TRs compared to lower TRs. This result confirms the finding from Fig 1C that the fish community in lower TRs ($\leq$10˚C, 10–15˚C, 15–20˚C) is more sensitive to the change of ST compared to higher TRs (20–25˚C, $\geq$25˚C). The increase of scaling exponents implies that the fish community experiences a significant change in species populations with impacts on the collective behavior (in terms of interactions) around 20˚C.

## 3.2 Interaction inference and temperature-dependent network characterization

Different patterns in epdf and Taylor's law of species abundances shown in Fig 3 envision the discrepancy in species interactions and dynamics of ecosystem for different TRs. In this study, species interaction is quantified by TE that is an information-theoretic variable measuring the amount of directed information flow and evaluating connectivity between species. Networks for the whole period and five TRs (see Fig 4) are therefore inferred by TE-based OIF model, and graphically visualized with *Gephi* [73]. To refine network structure, links whose interactions are lower than 0.01 are discarded by setting a threshold to filter TE. Optimal Information Flow (OIF) networks present different structural and functional properties for particular TR fish groups. Network for the whole period shown in Fig 4A provides a static overview of the fish community without considering the dynamical change of temperature or seasonality. It outlines causal relationships between species in the fish community, but is inadequate to tackle the dynamics of fish ecosystems driven by the fluctuation of temperature, and identify ecological and dynamical responses. Therefore, to understand how temperature affects the fish community in detail, networks for five TRs ($\leq$10˚C, 10–15˚C, 15–20˚C, 20–25˚C and $\geq$25˚C) are reconstructed and listed in Fig 4 (from Fig 4B to 4F). Here we first define network size as an indicator that is proportional to the number of nodes and the total amount of interactions between species. By looking at the structure of networks (Fig 4B–4F) and taking total interactions over temperature (S12F Fig) into consideration, it is obviously observed that network size is the largest for the fish community within a higher TR. Nodes connected by links with warm colors are regarded as a cluster in which species are significantly affecting others or affected by others. Network of 15–20˚C group shows a larger cluster with stronger interactions between species compared to other TRs. Yet, the distribution of species interactions in this TR presents more evenness considering the gradient of links' color in networks. This finding can be obviously observed by comparing Fig 4B'–4F' describing the phase mapping of interaction matrices. OIF-based interactions for 15–20˚C group are distributed more randomly. This randomness reveals that OIF-inferred network for 15–20˚C group seems to be less scale-free versus other networks, and stands on a critical point where the fish ecosystem is experiencing a phase transition from a stable state to a metastable state. When considering specific species within different TRs, except for the $\leq$10˚C and 10–15˚C groups in a globally stable state, species 6, 7, 8, 9 are always the nodes with warm color (high OTE) (see Fig 4A and 4C–4F) in networks. Yet, interactions between these species are higher than others considering the phase mapping of TE matrices shown in Fig 4.

We also set a threshold as top 20% TE considering the Pareto principle (also known as the "80/20 rule") which specifies that roughly 80 percent of the consequences come from 20 percent of the causes in many events [65]. TE threshold following this rule is generally higher than 0.01. It would further reduce network size, making inferred networks briefer relative to the networks shown in Fig 4. OIF networks (S6 Fig) for the whole time period and five TRs with the threshold of top 20% TE are reconstructed using the same regulation used in Fig 4. After the further reduction of network size, networks in S6 Fig ignore more functional details,

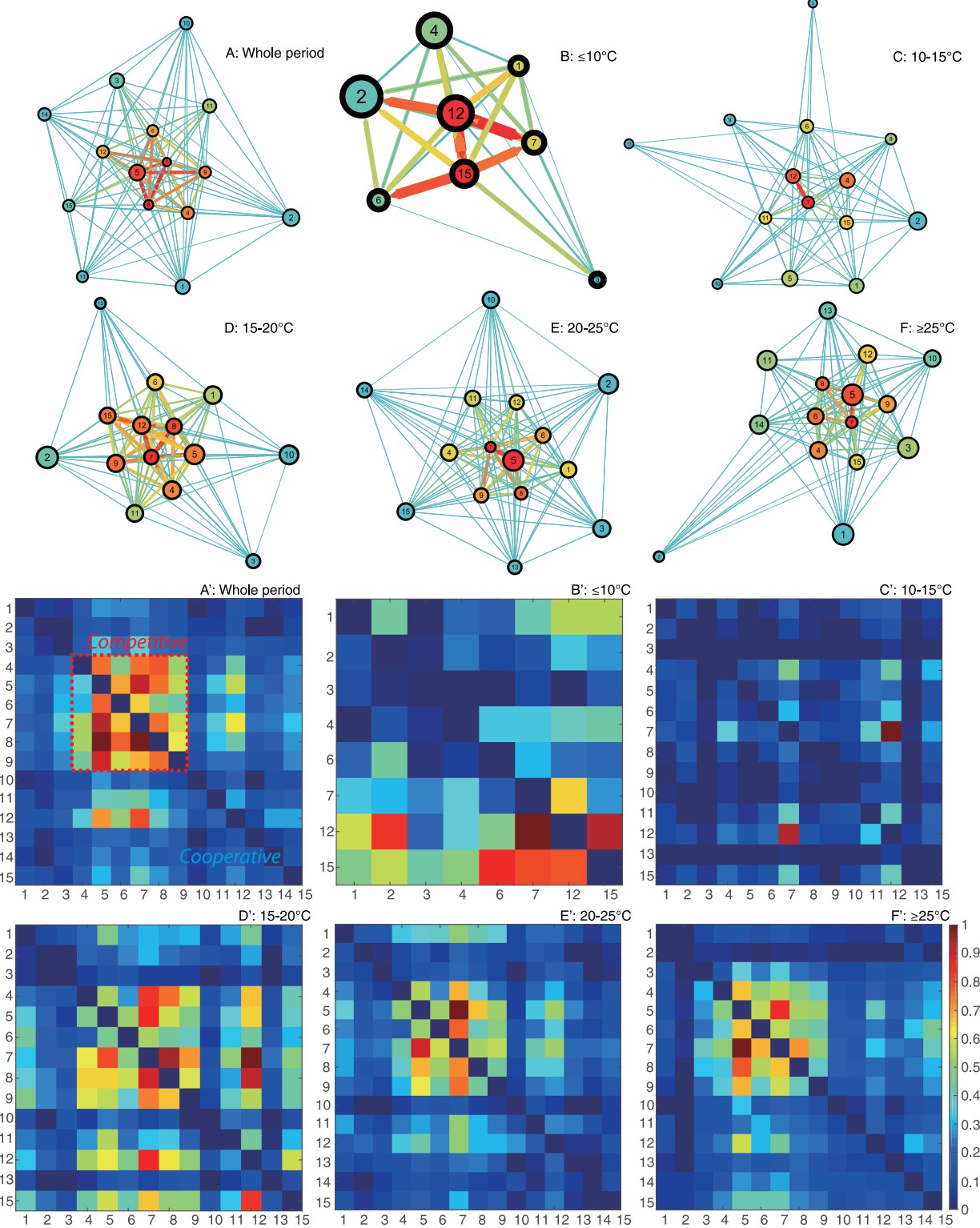

**Fig 4. OIF-inferred species interaction networks and matrices.** The OIF model is used to infer the causal interactions between all pairs of species, yielding average interaction (TE) matrices for the whole time period and five TRs. TE values in interaction matrices are normalized to 1 and drawn in plots A', B', C', D', E' and F'. After removing weak interactions by setting a threshold (TE = 0.01) to filter TEs, species interaction networks are reconstructed using *Gephi* (https://gephi.org/) and shown in plots A, B, C, D, E and F. The size and color of nodes is proportional to the Shannon Entropy of species abundance and the total outgoing transfer entropy (OTE) (the greater OTE, the warmer the color); the width and

color of the link between species are proportional to species interactions computed as TE (the greater TE, the warmer/wider the link's color width). Each network is plotted independently from the others so colors and sizes are relative to each TR range. The arrow of links defines the direction of species interaction (TE) that are bidirectional and asymmetrical. High/low values of TE define competitive/cooperative species dynamics that is associated to different network motifs (small-world and scale-free, respectively); increases in temperature over time leads to scale-free loss and increased high interactions with dominance of competitive species with summer-like dynamics.

but are clearer to identify the structural difference between networks compared to Fig 4. Furthermore, it is possible to statistically analyze the structural degree, in- and out-degree of nodes for each network. Statistical analysis for nodal degree is shown in S7 Fig. Structural degree, in- and out-degree get higher with the increase of temperature as a whole. This result implies that warmer conditions make the fish community more socially connected. Separately, pdf of structural degree (see S7A Fig) shows that structural degree of nodes increases with the increasing temperature, and favors bimodal distribution with different shapes for ≤10˚C, 10–15˚C, 15–20˚C, 20–25˚C groups, while roughly uniform distribution for ≥25˚C group. For ≤10˚C group, most structural degrees are distributed within the lower unimodal range from 1 to 3. With the increasing temperature, more and more values of structural degree are distributed in the higher unimodal range. However, for the highest TR (≥25˚C) group, the pdf of structural degree presents more evenness within the range from 0 to 10 (uniform distribution) compared to other groups. This result implies that structural degree does not increase continuously with ST, and that the fish community would become less interacting when ST is too high. Analogous features can be observed in the distribution of in-degree (S7B Fig) and out-degree (S7C Fig). For each TR, OTE and average abundance are calculated for each species without considering any threshold for TE. Species' OTE favors bimodal distribution with short range for ≤10˚C and 10–15˚C groups, and broad range for 15–20˚C, 20–25˚C and ≥25˚C groups (see S8A Fig). This result suggests that overall effects of species on others grow with the increasing temperature. In addition, OTE is scaling with mean abundance (a recurrent pattern known as Kleiber's law as found for other ecosystems [35]) and abundance standard deviation that is mass-independent (S11A and S11B Fig, respectively).

### 3.3 Interaction spectrum, phase transition and system stability

The probability distribution of species interactions is characterized by calculating discrete exceedance probability, then fitted using power-law function (see Fig 5). Among five TRs, the greatest interaction occurs in the highest TR (≥25˚C) group, indicating that species interactions overall increase with ST. The scaling exponent of the power-law distribution (computed as $1 - \varphi$) is lower for ≤10˚C, 10–15˚C, ≥25˚C groups and the first regime (black fitting) of 15–20˚C group compared to 20–25˚C group and the second regime (green fitting) of 15–20˚C group. The power-law model fits the epdf of TE better for ≤10˚C and 10–15˚C groups relative to other TRs. The better fitting demonstrates that the distribution of species interactions presents more scale-free patterns, implying a globally stable state for these two groups. With the rise of temperature, the fish community begins to get more interacting gradually and two regimes therefore appear in the distribution of species interactions for 15–20˚C group. The distribution of species interactions for 15–20˚C group is divided into two sections by a break point ($y_{min}$ = 0.584 that is higher than the median value of TE, 0.263) estimated from power-law fitting; yet, the scaling exponents of the two epdf regimes are very different (-0.357 and -4.468, respectively) identifying a critical and subcritical regime, respectively [49, 50]. The critical regime is more stable (optimality sensu) due to higher scale-free organization vs. the subcritical that is less optimal but more dynamically stable or resilient (a la' Scheffer). The two different regimes for the 15–20˚C range suggest that the fish community stands on a critical

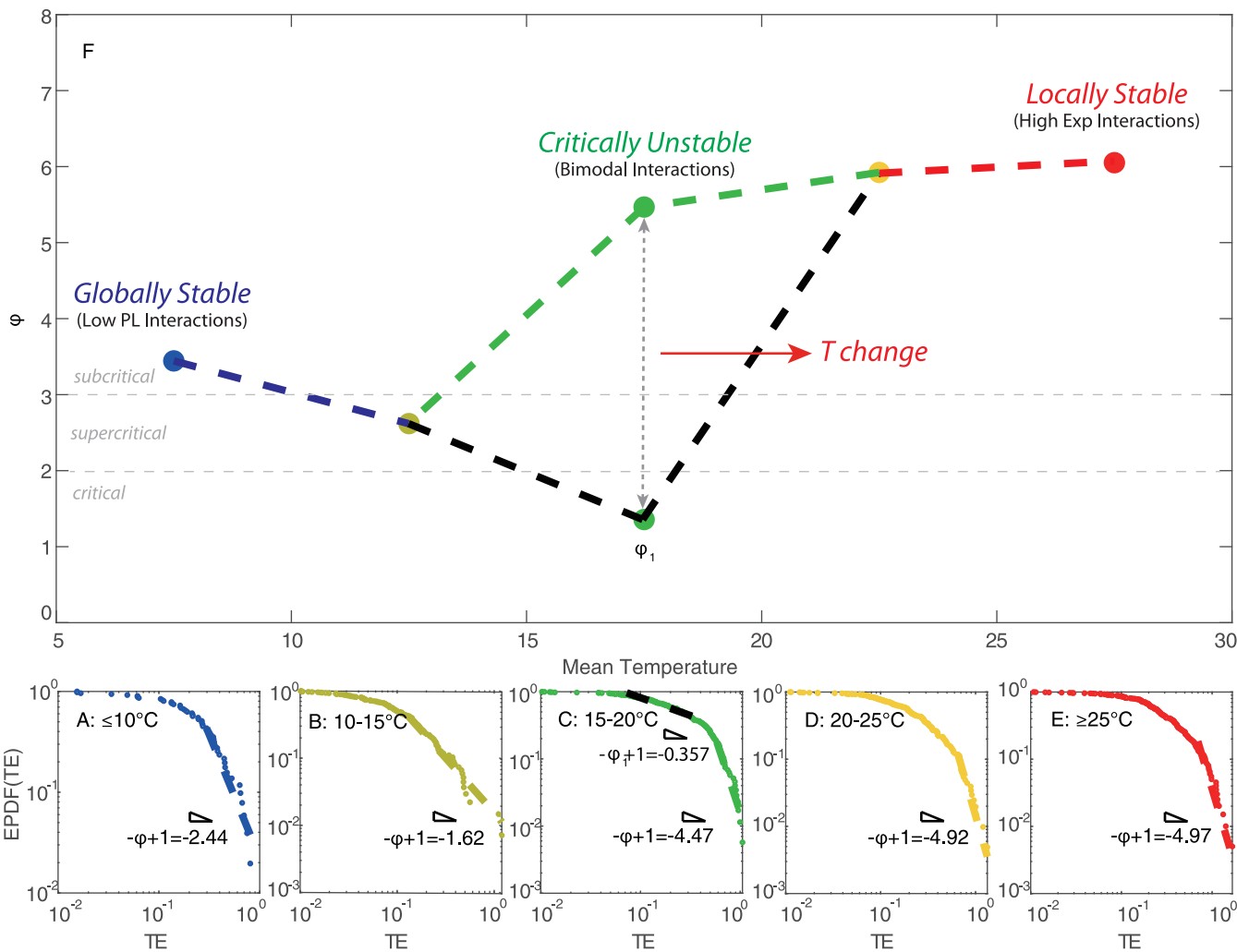

**Fig 5. Exceedance probability distribution function of species interactions and phase-space dependent on surface temperature. A**, **B**, **C**, **D** and **E**: Epdfs of OIF-based interactions (after filtering with a threshold of 0.01) between all pairs of species for five TRs: ≤10˚C, 10–15˚C, 15–20˚C, 20–25˚C, ≥25˚C. The fit is with a power-law function. In plot **F**, all power-law exponents are connected by dashed lines. Scaling exponents for different TRs are displayed in a phase-space where x-coordinates are middle values of TRs (note that 7.5˚C and 27.5˚C are selected as middle temperature values for TRs lower than 10˚C, and higher than 25˚C, respectively); points are empirically connected by a dashed line from low to high temperature to emphasize phase-transition of interactions. Note that for the 15–20˚C group, epdf of species interactions presents two power-law regimes that are separately fitted by a power-law functions. The double-Pareto distribution reflects the presence of two strange-attractors and a critical unstable transition at 15–20˚C. Critical regimes are defined according to power-law criteria defining finiteness of statistical moments (see [35]).

point where the fish ecosystem is experiencing a phase transition from global stability to meta-stability, and that the collective dynamics of species is experiencing a significant change driven by ST. The change of collective behavior is also reflected by the fluctuations of species populations and richness. The power-law of species interactions for 20–25˚C group shows the highest scaling exponent, and the distribution of species interactions tend to be the most exponential among five TRs. The power-law model was forced for the tail of the distribution in order to provide a phase-space plot as a function of power-law exponents; however, the maximum likelihood solution is an exponential distribution for TR 20–25˚C and above 25˚C. Species interactions continue to increase in magnitude as temperature rises, and a large number of weak interactions emerge simultaneously, leading to a distribution pattern where most interactions

are clustered from low to intermediate range forming a small-world network topology. This distribution pattern exhibits a less scale-free topology compared to other TRs. Therefore, fish ecosystems within 15–20˚C and 20–25˚C groups are metastable (also see Fig 4D, 4E, 4D' and 4E'). With the continuous increase of ST ($\geq$25˚C), the power-law scaling of the species interaction distribution shows a significant decrease that is opposite to ST. The power-law fitting model fits the epdf of species interactions better for $\geq$25˚C compared to the critical group of 15–20˚C and exponential group of 20–25˚C. This result indicates that several major species interactions continue to rise, while minor interactions disappear, leading to a flatter distribution and more scale-free pattern relative to 15–20˚C and 20–25˚C groups. The fish ecosystem returns to another relatively stable state after perturbations caused by ST. This finding confirms the fact that the fish community would not get more interacting continuously with the increase of ST, yet the activity of fish species would decrease when temperature is too high. The trend of scaling exponents (Fig 5F) shows that the fish ecosystem is subjected to a phase transition from global stability to metastability in 15–20˚C group, and finally returns to local stability (globally unstable Bak sensu, but resilient Scheffer sensu) with the increase of temperature. A fold-like bifurcation occurs considering the transition of species interaction at 15–20˚C. This result reveals that, during the phase transition from global stability to local stability, the fish ecosystem stands at a critical point within 15–20˚C where the fish community experiences a significant change in collective dynamics. These changes result in different functional features for the fish ecosystem (and potential environmental transitions, e.g. in biogeochemical cycling) within different TRs as graphically shown by functional interaction networks (see Fig 4).

In this work, the dominant eigenvalue of species interaction (TE) matrix is studied as a representation of network structure and to analyze the stability of the fish ecosystem. According to the above analysis of species interaction spectrum, OIF-inferred networks representing the fish community present different structural patterns for different TRs. As a part of the complex marine ecosystem, these structural differences may relate to numerous relevant biotic and abiotic factors. To prove ST to be an important factor affecting the structure and function of networks, we infer dynamical networks over time for each TR, then calculate the information flow from ST to the real part of the dominant eigenvalues of dynamical TE matrices underlying network stability (see Table 2). Table 1 reports various ecosystem metrics of species populations and community considering time variability of the environment. The information ow from ST to network structure increases from $\leq$10˚C to 20–25˚C group, then slightly declines. The lower information ows for $\leq$10˚C and 10–15˚C groups manifest less effect of ST on these TRs, implying more stable states with higher resilience to the fluctuation of ST compared to other groups. In contrast, 15–20˚C and 20–25˚C groups show high information ows from ST to network structure, implying strong effect of ST on the fish community. This

**Table 2. Predicted causality between temperature and ecosystem stability for different temperature groups.** The potential effect of temperature variability over time on ecosystem stability is calculate as Transfer Entropy (TE) between temperature (within TR groups) and the real part of the dominant eigenvalue (see S12A Fig for the latter). The larger TE the larger the predictive causality.

| TRs | TE($T \rightarrow Re\{\xi_{max}(t)\}$) |
|---|---|
| $\leq$10˚C | 0.50 |
| 10–15˚C | 0.99 |
| 15–20˚C | 1.17 |
| 20–25˚C | 1.26 |
| $\geq$25˚C | 1.06 |

corresponds to the relatively unstable state (metastability) where these groups undergo a significant change in system state and dynamics. The slight decrease in information ow from 15–20˚C and 20–25˚C groups to ≥25˚C group means the reduction in the effect of ST on (the improvement of system stability for) ≥25˚C group. These results extraordinarily conform to the findings from the probabilistic analysis of species interactions for TRs. Therefore, the change of ST is considered as an important factor that drives structural changes and state shifts of the fish ecosystem.

Eigenvalues of species interaction matrices for each TR are scattered together in a complex plane (Fig 6). This method displays the position of all eigenvalues in the form of an ellipse [22, 68]. There are 8 eigenvalues for ≤10˚C group, 14 eigenvalues for 10–15˚C group, 14 eigenvalues for 15–20˚C group, 15 eigenvalues for 20–25˚C group and 15 eigenvalues for ≥25˚C group. Eigenvalues correlate with the stability and resilience of the fish ecosystem in both magnitude and variance. When only considering the eigenvalues with positive real parts that may result in instability, there are 2 eigenvalues with positive real parts (2.028 and 0.009) for ≤10˚C group, 5 eigenvalues with positive real parts (2.262, 0.002, 0.002, 0.003 and 0.0003) for 10–15˚C group, 1 eigenvalue with a positive real part (4.998) for 15–20˚C group, 1 eigenvalue with a positive real part (4.934) for 20–25˚C group and 2 eigenvalues with positive real parts (6.143 and 0.052) for ≥25˚C group. It is observed that each TR has eigenvalues with positive real parts that may bring instability to the ecosystem, while positive real parts are greater for higher TRs (especially for 15–20˚C, 20–25˚C and ≥25˚C groups). Sea temperature is quite likely the strong causal factor. This result confirms the finding that the fish ecosystem presents more instability for 15–20˚C, 20–25˚C and ≥25˚C TRs.

## 3.4 Network-based ecological importance and critical species identification

Biological importance and critical species identification are performed by further analyzing structural features of OIF networks and combining interaction spectrum with abundance spectrum. To this end, we first measure the salience for all links in OIF networks [69]. Link salience computation is based on the effective proximity $d_{ij}$ defined by the strength of mutual interactions: $d_{ij} = 1/TE_{ij}$. It is intuitively assumed that strongly (weakly) interacting nodes are close to (distant from) each other. Link salience matrices shown in Fig 7 illustrate that species 7 (*Pseudolabrus sieboldi*) is always the reference node of the most salient links with the exception of ≤10˚C group where the reference node of the most salient links is species 15 (*Rudarius ercodes*) instead of species 7. According to the website of *fishbase* (http://www.fishbase.org/summary/Pseudolabrus-sieboldi.html), species 7 (*Pseudolabrus sieboldi*) is a native species in northwest pacific ocean and mainly distributed in Japan waters [74]. These results reveal that native species 7 has the most critical influence on other species in the fish community, and plays a vital role in maintaining structure and function of networks.

We also calculate Shannon entropy for species to measure the fluctuation and uncertainty of species abundance, and compute species' OTE as another nodal (species) property to quantify how much a species totally influences others in OIF networks. Then, species rankings considering top 5 Shannon entropy and top 5 OTE are listed and shown in S9 Fig. On the one hand, the species ranking of Shannon entropy (left figures in S9 Fig) shows that species 2 or (and) 5 always have the greatest Shannon entropy that represents the highest uncertainty in abundance for these two species. This finding also can be observed in the original time series of abundance plotted in S3 Fig. Species 2 and 5 have the highest fluctuations and diverse values of abundance. On the other hand, the species ranking of OTE (right figures in S9 Fig) shows that for all TRs except the ≤10˚C group, species 7 always has the greatest OTE, and species 5, 6, 7, 8 and 9 are often involved in top 5 OTE ranking especially for higher TRs. These species

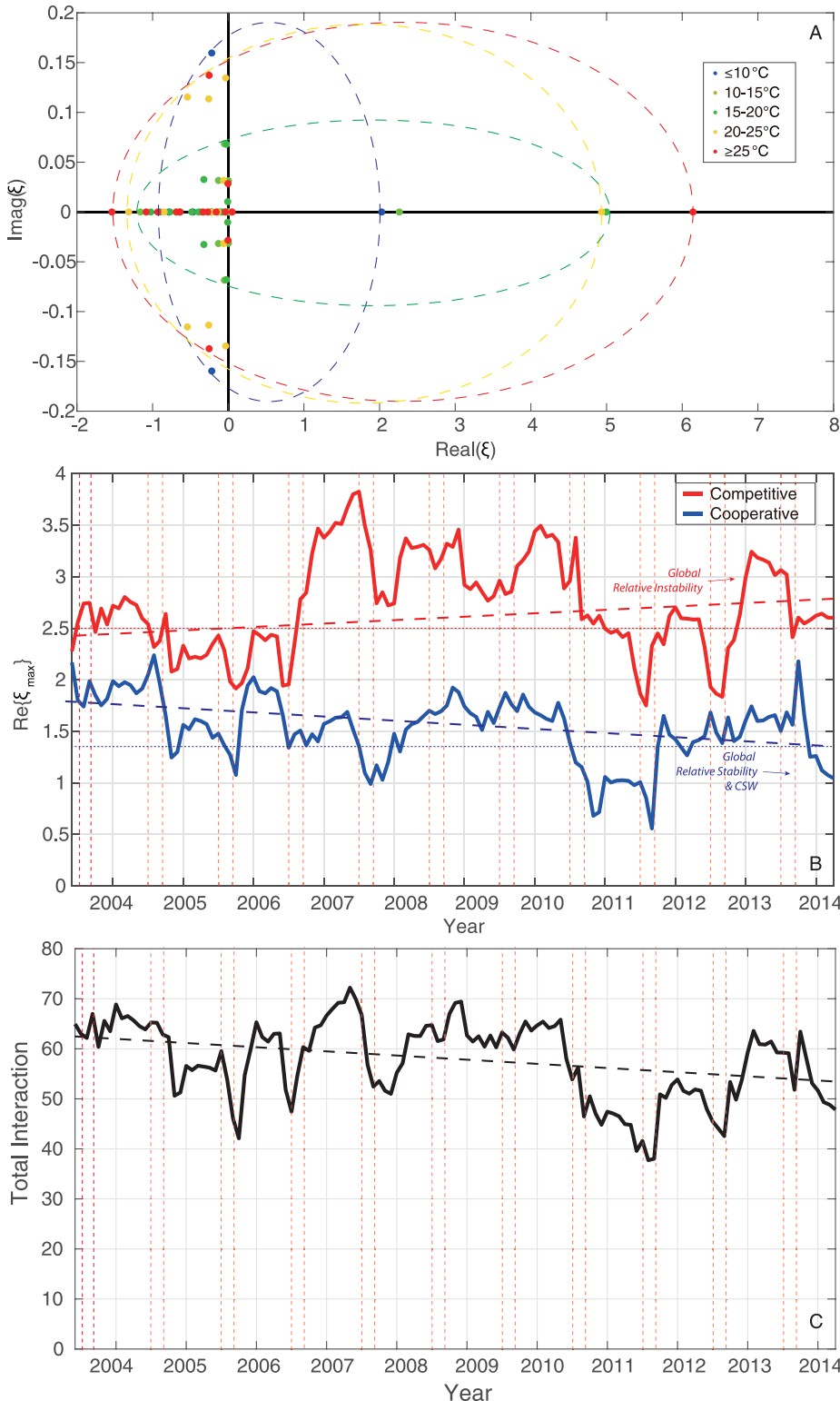

**Fig 6. Eigenvalue spectrum of TE interaction matrices across temperature ranges and ecosystem relative stability over time.** Eigenvalue distribution of TE interaction matrices are scattered in a complex plane (five different colors correspond to five TRs) (**A**). Dynamical stability of the fish community is computed as the real part of the dominant eigenvalue of TE interaction matrices; this is show over time (**B**) for competitive (4–9 species) and cooperative species. The total interaction of dynamical networks over time (as sum of all TE interactions) (**C**) is mirroring the fluctuations

of the eigenvalue relative stability. Note that fluctuations of the eigenvalue and interactions are increasing over time and manifesting critical slowing down (CSD) toward a suboptimal ecosystem state with small-world/random high interactions (Fig 5) with large fluctuations in abundance synchronized with temperature (Figs 1 and 3). Total interaction dynamics is dominated by cooperative species but extremes are regulated by competitive species that are much more synchronized with ST fluctuations. This is because cooperative species are more numerous and abundant than competitive species and the latter are synchronized with temperature extremes.

are dynamically competitive species a' la [28] and with higher temperature fitness as well as smaller geographic range tendentially. For instance species 5 *Trachurus japonicus* (Horse mackerel) in S15 Fig is native of Japan while species 9 *Halichoeres tenuispinnis* (Chinese wrasse) despite being mostly distributed in Japan is invasive with origin in SE China. These finding imply that these competitive species have significant effects on other species in the fish community, of which species 7 is the greatest. This result is in accordance with the results of link salience matrices shown in Fig 7. In a word, through measuring link salience and OTE, species 5, 6, 7, 8, 9 can be identified as the critical species in the fish community. Additionally, we also calculate TE, MI, species abundance vs. ST in semi-log scale, Pearson correlation coefficient (indicated as cc) and causality coefficient $\rho$ of CCM [63] (Table 3) between species

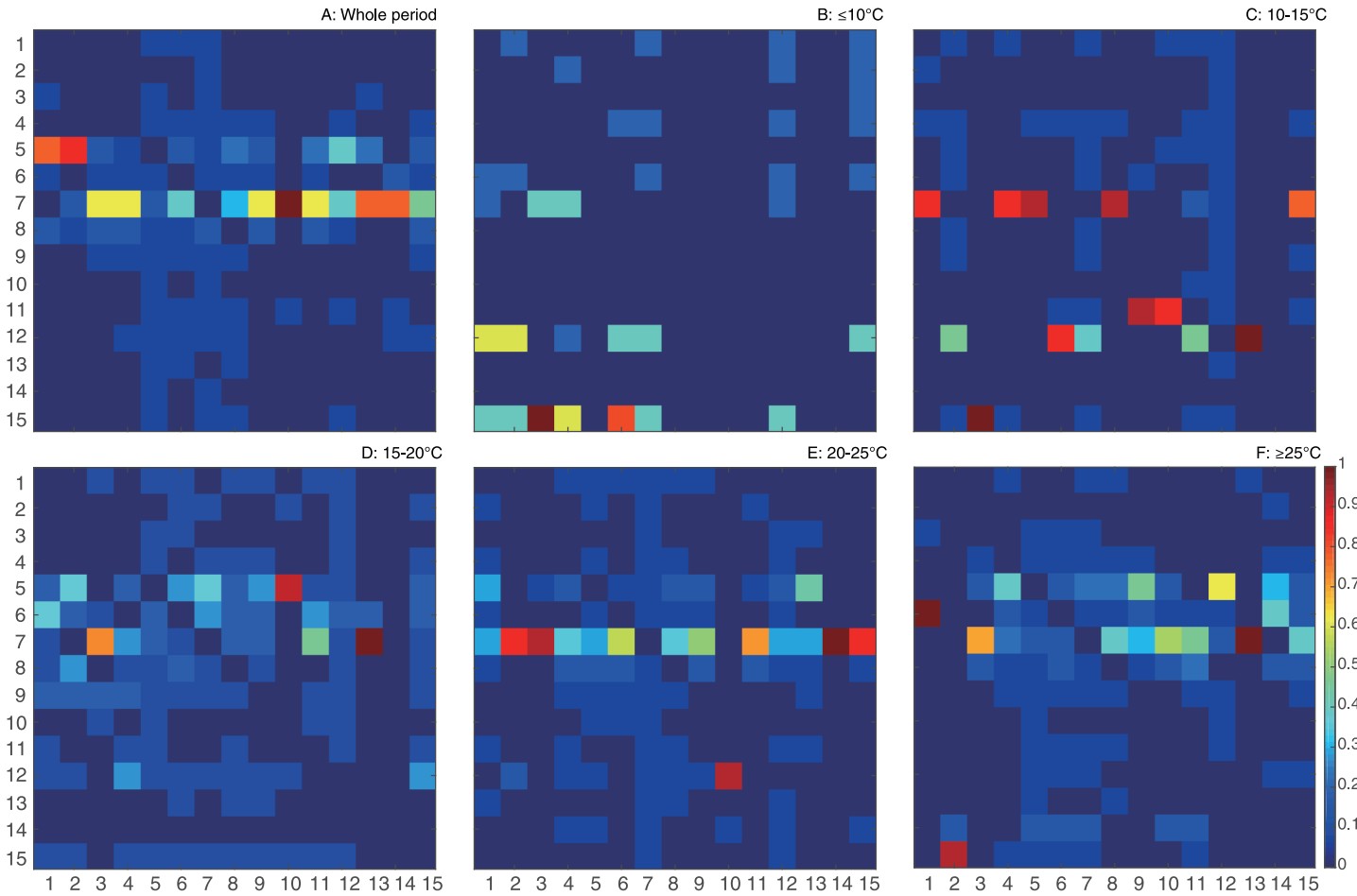

**Fig 7. Link salience matrices highlighting Pareto links.** Link importance is measured as link salience for all OIF networks corresponding to the whole period and five TRs, yielding 15 × 15 matrices. The values of link salience are normalized to 1, and drawn in plots A, B, C, D, E and F, respectively. Pareto links are between keystone species responsible for the stability of the ecosystem.

**Table 3. Indices measuring the relationship between ST and species abundance.**

| Species | TE($T \to s_i$) | TE($s_i \to T$) | MI | slope | cc | $\rho$ |
|---|---|---|---|---|---|---|
| 1. *Aurelia aurita* (Moon jellyfish) | 0.023 | 0.005 | 0.316 | 0.137 | 0.157 | 0.095 |
| 2. *Engraulis japonicus* (Japanese anchovy) | 0.001 | 0.014 | 0.132 | -0.082 | -0.066 | 0.290 |
| 3. *Plotosus lineatus japonicus* (Sea catfish) | 0.029 | 0.002 | 0.198 | 0.277 | 0.289 | 0.310 |
| 4. *Sebastes inermis* (Black snapper) | 0.014 | 0.031 | 0.411 | 0.092 | 0.278 | 0.318 |
| 5. *Trachurus japonicus* (Horse mackerel) | 0.171 | 0.014 | 0.620 | 0.066 | 0.615 | 0.624 |
| 6. *Girella punctata* (Blackeye seabream) | 0.107 | 0.000 | 0.472 | 0.129 | 0.476 | 0.385 |
| 7. *Pseudolabrus sieboldi* (Wrasse) | 0.071 | 0.044 | 0.691 | 0.112 | 0.712 | 0.690 |
| 8. *Halichoeres poecilopterus* (Rainbow wrasse) | 0.021 | 0.007 | 0.712 | 0.043 | 0.506 | 0.447 |
| 9. *Halichoeres tenuispinnis* (Chinese wrasse) | 0.007 | 0.006 | 0.707 | 0.099 | 0.501 | 0.424 |
| 10. *Chaenogobius gulosus* (Goby) | 0.002 | 0.009 | 0.127 | 0.119 | 0.105 | 0.063 |
| 11. *Pterogobius zonoleucus* (Blue/Yellow striped Goby) | 0.004 | 0.001 | 0.276 | 0.166 | 0.258 | 0.120 |
| 12. *Tridentiger trigonocephalus* (Chameleon Goby) | 0.010 | 0.024 | 0.264 | -0.016 | 0.151 | 0.316 |
| 13. *Siganus fuscescens* (Rabbitfish) | 0.026 | 0.010 | 0.207 | 0.250 | 0.243 | 0.294 |
| 14. *Sphyraena pinguis* (Red barracuda) | 0.039 | 0.005 | 0.194 | -0.012 | 0.229 | 0.199 |
| 15. *Rudarius ercodes* (Pigmy filefish) | 0.023 | 0.015 | 0.389 | 0.079 | 0.237 | 0.227 |

abundance and ST to identify which species are most affected by ST. The identification is done by comparing the aforementioned linear and non-linear methods that were also used to compare inferred species interactions (see S5 Fig, and [28]). The slope of the abundance-temperature relationship in semi-log scale is the parameter of the exponential function of species abundance growth dependent on ST (see S4 Fig). In Table 3, TE from ST to species abundance for species 5, 6 and 7 is higher than other species. MI, Pearson correlation coefficient and $\rho$ are overall higher for species 5, 6, 7, 8 and 9. Therefore, species 5, 6, 7, 8 and 9 are considered as the species that are most affected by ST; this is not necessarily a negative impact of temperature but rather a signature of importance of temperature in constituting the ecological niche of these species. It is interesting that these species are also recognized as critical species in the fish community by measuring link salience (dependent on TE) and OTE ranking of importance for predicting collective dynamics. These competitive species (dynamically speaking as in [28]) are highly concordant with each other and more synchronized with temperature, yet more likely affected by temperature fluctuations directly. We also find that TE values from species abundance to ST are very small. This result is expected since fish species do not affect ocean temperature. By looking into S4 Fig, the abundance of species 5, 6, and 7 exponentially grows with ST more obviously than other competitive species. Competitive species have the strongest and positive exponential fit with seasonal temperature versus cooperative species whose response to seasonal temperature is less exponential and more frequently negative (implying a decay); the exception is for species 15 that is dynamically the most similar to competitive species. Competitive species (dynamically speaking as in [28]) are highly concordant with each other and more synchronized with temperature, yet more likely affected by temperature fluctuations directly. The exponential fit is not by chance a by-product of the synchronized exponential fluctuations of ST and fish abundance, while cooperative species fluctuations are more power-law distributed (S2 Fig) (although in terms of range than critical regime). This signifies that cooperative species are affected also by other factors such as fishing beyond linkages with competitive species-ST.

Moreover, probability distribution functions of abundance for all species (see S2 Fig) show that species abundances of these species are more evenly distributed within relatively small

ranges (the least range is [0,45) for species 7) compared to other species with wide distributions and extreme values. It demonstrates that trajectories of species abundance for species 6, 7, 8 and 9 are remarkably divergent from other species. Species with wider distributions and more extreme values present more uncertainty in abundance, that is, Shannon entropy of these species is higher than that of species 6, 7, 8, 9. Intuitively, TE-s from source variables (species) with low uncertainty (more deterministic information) in abundance (species 6, 7, 8, 9, for instance) to target variables with high uncertainty (see left figures in S9 Fig) are supposed to be high. Considering species 6, 7, 8 and 9 themselves, within the small range of abundance distribution, relatively speaking, these species also present transparent divergences (dissimilarity) in the distribution of species abundance. Therefore, causal interactions (transfer entropy) between these species are relatively high. These results address that the distribution of species abundance can be used as a proxy of interactions by comparing similarity and divergence. Take species 2 (*Engraulis japonicus*) as an extreme example, TE-s from species 2 to others are relatively low as expected since the original abundance data of species 2 have many zeros or values close to zero that lead to lower uncertainty despite the asynchrony and divergence.

Considering the whole time period, rather than solely estimate the simple continuous pdf of abundance for each species (S2 Fig), we probabilistically characterize distributions of abundance by computing epdf and using power-law fitting. The scaling exponent of power-law fitting is considered as another species property in terms of the distribution of abundance (see S1 Fig). Thereafter, abundance-interaction phase-space describing relationships between power-law scaling and species OTE (power-law exponents vs. OTE for all species, indeed) is shown in Fig 8.

For a selected species, the higher the exponent of the power-law, the more abundance values are distributed within a narrower range (S1 and S2 Figs); additionally, the higher OTE, the more strongly the species interacts with others.

The phase-space mapping in Fig 8 shows a trend where the power-law exponent of species abundance is proportional to the species OTE with species 3 and 6 as outliers. This result suggests that species whose abundance has a wide distribution with a fat tail have relatively low total effects on other species (these are cooperative species, while species whose abundance is more evenly distributed within a narrow range have high total effects on other species (species 7, 8, and 9 but in general all competitive species). These findings, that highlight the duality between abundance and interaction distribution, are also extractable from the mapping of interactions for the whole time period shown in Fig 4A'. Previous research showed that species with more exponential distribution of abundance have a competitive role in the fish community dynamics [28]; this is beneficial for species abundance growth and regulation of other species. The former are acclimatized species where temperature affect them positively with higher ecological memory while the latter are affected by multiple environmental pressure such as temperature and fishing (temporally and likely spatially in the geographical area considered) and are likely more sensitive to short-term conditions. In S9 Fig, OTE vs. mean abundance shows that for most species, effects on other fishes are proportional to the abundance of species (S9A Fig) (this is known as Kleiber's law [35]), while inversely proportional to the standard deviation of the abundance of species (S9B Fig).

## 3.5 Time- and temperature-dependent dynamical interactions and stability

Considering both dimensions of time and temperature, time- and temperature-dependent dynamical species interaction (TE) matrices are inferred from time series of each time and temperature unit, respectively. On the temporal scale, S12A Fig shows the real part of the dominant eigenvalue of TE matrix (blue line) and adjacency matrix (red line) underlying the

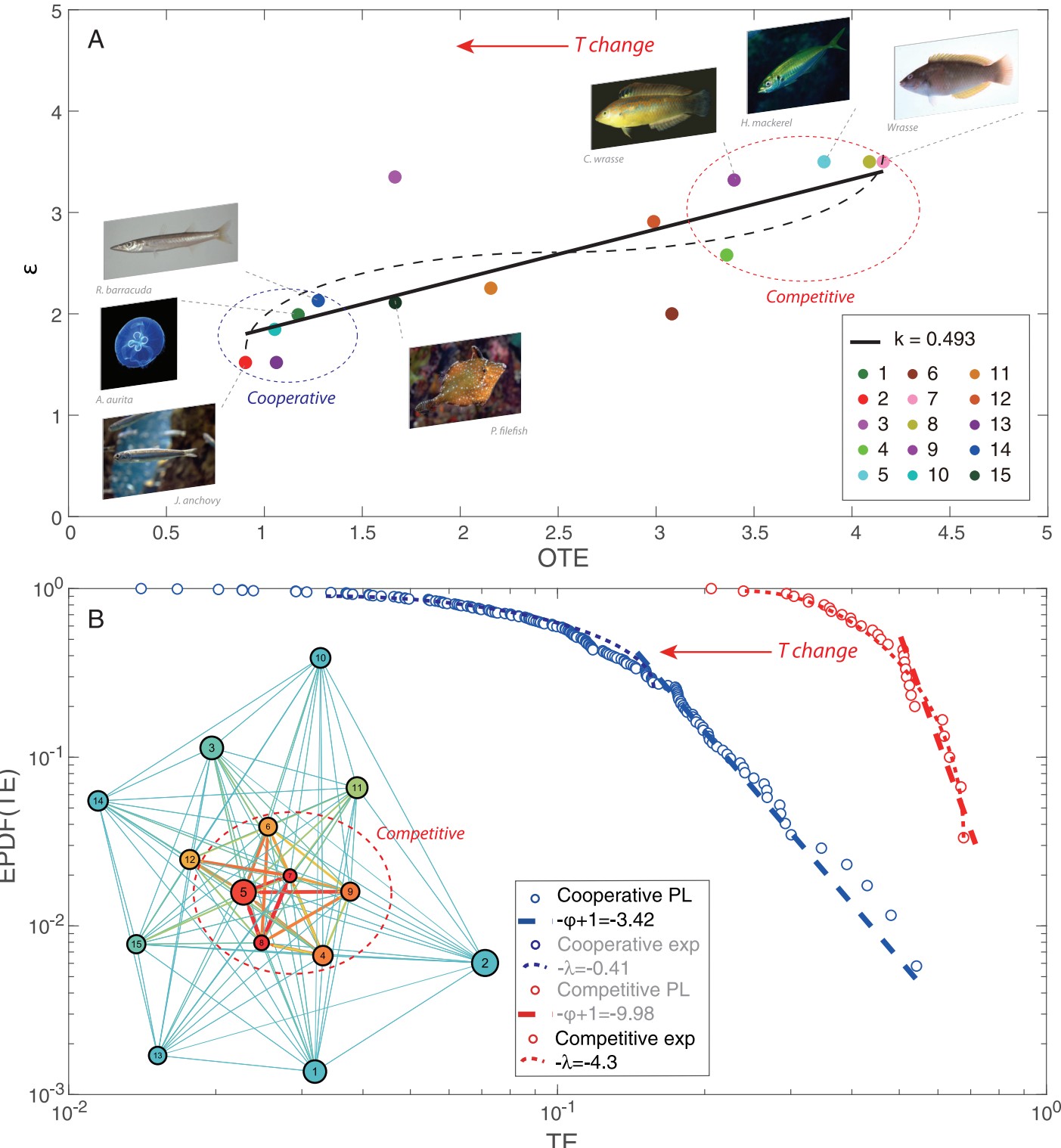

**Fig 8. Abundance-interaction atemporal pattern and distribution of species interactions for competitive and cooperative species.** (A) Considering the whole time period for the 15 species, ε is the slope of the Zipf's law (on pdf) of species abundance (see Fig 3) and OTE is the sum of all interactions of a species (that is how much one species influences the collective on average). Black continuous and dashed curves are the linear and non-linear (second-degree polynomial) fitting for the abundance-interaction pattern. Cooperative species, asynchronized with temperature, are characterized by more power-law distributed fluctuations in abundance (see S1, S3 and S4 Figs) manifested by the lower ε and wide scale-free distributed total interactions (Figs 4 and 5, and 8B) (with relatively low average) that are more stable over time (Fig 6); this underlines the weak-interaction induced stability of ecosystems. Increasing temperature leads to a predominance of small-world distributed high

interactions of competitive species (whose abundance is synchronized with temperature and distributed more exponentially); this causes the loss (or a shift) of Pareto salient links (Fig 7) and the higher fitness of invasive species with "hybrid" dynamics. (B) Epdf of interactions of competitive and cooperative species. Interactions are estimated via the OIF model for the whole time period. The number of cooperative species is higher and their scale-free distribution wider than competitive species as well as much more stable due to the lower power-law exponent; this is when considering the extreme competitive interactions as power-law although the whole distribution is exponential ($P(Y \geq y) \sim e^{-\lambda y}$). The rarity of competitive species is manifested by the narrower distribution with lower stability (Bak sensu). The legend shows in black the dominant dynamical regime for competitive and cooperative species (for the former only the exponential dynamics is substantiated); the inset is the species interaction network for the whole period (Fig 4A) with the competitive core highlighted.

structure of temporal networks. The real part of the dominant eigenvalue of TE matrix over time is lower than that of adjacency matrix, yet the former presents an obvious seasonal fluctuation. In the first half-year period (approx. from winter to early summer), it is observed that the real part of the dominant eigenvalue increases over time and reaches a spike during this period. Then it decreases in the second half-year period, while increases again at the end of year. The increasing trend or the high level of the real part of the dominant eigenvalue always appears within the time period with higher changes (fluctuations) of ST (from April to June or from October to December) in a year which probably corresponds to 15–20°C to 20–25°C TRs. These results suggest that the fluctuation of species interactions becomes higher and fluctuates more frequently during the time period with higher fluctuations of ST, and that the fish ecosystem tends to be metastable as the fish community becomes active. For adjacency matrix, the high level of the real part of the eigenvalue appears synchronously with that of TE matrix, while it is hard to identify the seasonality due to lower fluctuations. S12B Fig shows the magnitude of total interactions over time that is computed as the sum of TE values in interaction matrices of temporal networks. Total interaction over time presents clear seasonal fluctuations that highly synchronize with the seasonality observed in eigenvalues of TE matrix shown in S12A Fig (blue line). This finding verifies the result that species interactions in the fish community increase during the time period with high fluctuations of ST. S13 Fig shows how the dominant eigenvalue is the largest for intermediate and high TR as much as the functional network degree. Furthermore, species OTE is calculated for each temporal network, obtaining 256 OTE-s for each species. Pdfs of species OTE show that critical species 6, 7, 8, 9 have higher effects on other species in the fish community compared to others (see S14 Fig).

Considering temperature, S12E Fig shows how the real part of the dominant eigenvalue of TE matrix (blue line) and adjacency matrix (red line) underlying the structure of temperature-dependent networks changes with the increase of temperature, respectively. Both curves show that the real part of the dominant eigenvalue rises with the increasing temperature as a whole, while that of TE matrix has more informative fluctuations with higher frequency in contrast to the adjacency matrix. S12F Fig shows the magnitude of total interactions over temperature stemmed from TE matrices of temperature-dependent networks.

It is clearly observed that total interactions fluctuate analogously to the real part of the dominant eigenvalue of interactions (S12E Fig), as well as to the estimated effective $\alpha$-diversity derived from dynamical top 20% TE interactions (S12H Fig). These results indicate that interactions between species in the fish community are more sensitive to fluctuations of ST compared to macroecological $\alpha$-diversity that is a byproduct of interactions. Fluctuations of species interactions provide a potent indicator to monitor and predict ecosystem dynamics especially as a function of changes due to environmental stress. The effective $\alpha$-diversity in temporal and temperature-dependent dimensions is shown in S12C, S12D, S12G and S12H Fig.

## 4 Discussion

Growth, reproduction and living habits of organisms in the ocean, as well as external environmental factors including ST and daytime present a transparent seasonality. Therein, ST is

often perceived as a dominant determinant that drives the biological behavior of organisms in marine ecosystems [75–77]. Sea temperature has been affected by ocean warming caused by both global climate change and local anthropogenic stress. Ocean warming determined significant negative impacts on local fish communities and the whole marine ecosystem. In this paper, we first study the multispecies fish community in Maizuru bay by conducting abundance, macroecological and dynamical analyses over time. Particularly, these analyses are also conducted considering five TRs to investigate how ST affects fish community abundance and interactions. Temperature-dependent analyses capture the difference in macroecological indicators of the fish community among five TRs. Same analyses for EP, FS, native and invasive groups and two species (species 1 and 2) are performed to identify important trend in species groups. These analyses capture the seasonal fluctuations of ST, species populations and diversity, and the results support a speculation that the change of biodiversity in the fish community can be a biological response to the increasing fluctuations of ST. The species abundance of fish groups shows an overall slight increase since 2007 that is related to ST. Take 2009 as an example, the highest total abundance of species in EP, FS and native groups in this year is likely owing to the least fluctuation of ST. The increase of species abundance may improve species competition in the fish community. Those species whose abundances decrease with the increasing ST are supposed to compete with other species and present disadvantage in species interactions. This competitive disadvantage may lead to departure and extinction of some species, resulting in the global decrease of species diversity in the fish community. These valuable results are obtained from the analysis considering species abundance and diversity, while it is not sufficient to completely display the impacts of ST on the fish community, and limited to describe internal mechanisms of how ST affects the dynamics of the fish ecosystem.

To better understand biological responses of the fish community to the fluctuations of ST, the proposed information-theoretic OIF is employed to study the fish community considering species interactions and their dynamics over time. In information theory that has attracted considerable attention in complex networks, entropy (information) is used to quantify the uncertainty of a variable and its computation is based on the probability distribution of data. Transfer entropy is an asymmetric variable that measures directed relationships between two random variables by estimating the directed information flow between variables. In this study, time-series observations of species abundance, as research materials, are used by OIF to infer causal networks for the fish community. Information flow is therefore interpreted as the transfer of information about species abundance between species. The amount of the information flow quantifies how much one species can help predict another species in terms of species abundance [28]. In addition, information flow can represent the information transfer between other indicators in ecosystems dependent on research subjects and objectives, for instance, energy transferred between organisms in food webs, material flow through ecosystems, metabolism expenditure and net primary production.

The fundamental work of studying ecosystems via complex networks is to detect complex interdependencies comprised of a large number of causal interactions between species. The intuitive and heuristic notion in information theory for species interaction detection is transfer entropy. Therefore, OIF model is deployed in this study to detect potential species interactions that form information flow networks to model the fish ecosystem. Networks inferred from OIF model illustrate differences in structure and function among five TRs. These structural and functional differences imply different system states and dynamics the fish ecosystem possesses within different TRs. For ≤10°C and 10–15°C groups, the fish ecosystem lies on a globally stable state, then presents a shift of system state from stability to metastability in intermediate TRs, and finally returns to a locally stable state. Fish community on global stability is in the situation that the fish ecosystem is highly resistant to long-term perturbations

(change of species composition, ST, for instance) [78, 79], and intra- and inter-specific interactions are not too strong [80]. The newly established stable state is interpreted as locally stable state which addresses the fish ecosystem only resistant to small short-lived disturbances (minor fluctuations of ST, for instance) [81]. As well, different patterns are recognized for internal mechanisms of the fish ecosystem within different TRs by probabilistically characterizing species interactions. These shifts of ecosystem state and dynamics explain how ST substantially affects the fish community. In addition, eigenvalues of the interaction matrix are computed as indices to assess the ecosystem stability of the fish community [22, 68]. The determination of eigenvalues and eigenvectors of a system is extremely important to many problems in physics and engineering including stability analysis, oscillations of vibrating systems, to name only a few. It provides three measures of stability in terms of species interactions: (i) whether the fish community will return to the previous state after a certain perturbation, (ii) how fast the return will occur, and (iii) what the interactome dynamics of the fish community look like during the process of the return.

It is important to note that different species may respond to fluctuations of ST in different ways. As one of the many causal factors, temperature may lead to different species responses that can also improve the stability of fish ecosystems; some fish species for instance may exponentially grow as a function of ST (e.g. due to acclimatization or due to their higher fitness for the new dominant TR) and shift the ecological niche of the community. Yet, to further tackle the complexity ecosystem dynamics and identify the importance of some critical species under the conditions of ocean warming, species-specific analyses are conducted via network-based analyses to define which critical keystone species underpin shifts and stability.

Link salience is assessed as one of the network metrics to classify salient pairs in the fish community based on an importance estimate (consensus a' la [69]) of those pairs for of all nodes. Native species 7 (*Pseudolabrus sieboldi*) is the most frequently observed species in the salient links as a reference node. This result means that species 7 is likely to play a more significant role in maintaining the fish ecosystem stable compared to other species. Secondly, OTE that counts the total effects of one species on others is calculated for each species as a nodal property in networks. TE and OTE are considered as the strong indicators of ecosystem dynamics for pairwise and nodal functional characterization since those variables are related to species interdependence in the view of complex networks [35]. These information-theoretic indicators based on species interdependencies rather than species abundance and macroecological indicators lead to more informative ecological information and offer better characterization and understanding for the dynamical evolution of the fish ecosystem. In this study, OIF-inferred networks are predictive interaction networks. They can be used, given source species, to measure the uncertainty reduction of target species. Cause-effect relationships between two species are asymmetrical and the asymmetry is captured by the model. OTE is a variable measuring how much one species contributes to the prediction of next states of the fish ecosystem. From this framework, species 6, 7, 8 and 9 are identified as critical species often involved in top 5 OTE ranking, and species 7 has the greatest OTE for all TRs, except for the ≤10°C group. The results from OTE coincides with those from salient link assessment. This finding means that species 6, 7, 8 and 9 are core species in the fish community that are interacting strongly with each other, present quite different behavior compared to other species, and play a vital role in maintaining the stability of the ecosystem. More importantly, these species are native species in Maizuru bay (see S1 Table). The abundance of species 6, 7, 8 and 9 presents remarkable divergence from other species (see S2 and S3 Figs).

We also develop dynamical networks dependent on time and temperature. Relationships between eigenvalues (real part of the dominant eigenvalue of TE matrices as in [21, 22]) and system stability, species interactions (total interaction and interaction topology) and effective

diversity indicators (effective $\alpha$-diversity) are discussed. Dominant eigenvalue provides information on local stability which is a specific equilibrium point the system is around after perturbations and will return to on its own in some time. Its fluctuations over time reveal less stability for the fish ecosystem during the period with higher fluctuations of ST in one year. It means that given a specific stable state (equilibrium point), the ecosystem within this period is easier to oscillate around the equilibrium point and may take more time to recover. This result is in line with the relatively unstable state of 15–20˚C and 20–25˚C groups concluded in section 3.3, since these two TRs are more likely to appear in spring and autumn which have relatively high temperature fluctuations in one year. Dominant eigenvalues over temperature also suggest that in a certain TR ($\leq$18˚C), the increasing temperature rapidly makes the fish ecosystem more interacting, but less stable. Total interaction of dynamical networks present equivalent fluctuations to eigenvalues on both scales that explicitly imply the relationship between species interaction and system stability. Moreover, effective $\alpha$-diversity from OIF networks is able to roughly track the real taxonomic diversity. Therefore, dynamical network analyses provide a real-time and temperature-dependent monitor that could directly display the system states and species activeness in view of system dynamics. This is important and useful to understand how fish ecosystems respond to the change of ST, especially in conditions of ocean warming, in order to preserve their function along sustainable targets.

## 5 Conclusions

Ocean temperature is the dominant environmental factor shaping marine ecosystems, and specifically affecting fish species niche by modulating their collective behavior (interactions). Increasing trend and abnormal fluctuations of ST caused by global climate change, and amplified by local habitat pressure such as biogechemical loads and increased fishing, have created significant and likely irreversible challenges for fish ecosystems in the last 50 years [82]. In this study, the fish community in Maizuru Bay, Japan, is studied along time and temperature dimnewnsions considering patterns of interrelationships of community abundance, species interactions, taxonomic and effective diversity, and interaction network species salience. The following results are worth mentioning.

- 12-year species abundance and taxonomic $\alpha$-diversity time series present increased seasonal fluctuations associated to ST mean and variance increase. Critical slowing down (a' la Scheffer [38]) is manifested by the decreasing trend of abundance, especially for highly commercially valuable species, larger fluctuations in interactions, and dominant eigenvalue approaching zero defining frequent transitions. This lead to a more homogenous state dominated by competitive species with low or non-commercial value as well as invasive species increase with higher synchronicity with temperature (yet likely suitability) to occupy the climate niche of cooperative species. We use a previously developed Optimal Information Flow model [28] (validated on the same dataset) to infer species interaction networks for the period 2002–2014 and for each temperature group. The dominant eigenvalue, derived from a stability analysis of in the ecosystem interaction matrix, is mildly decreasing over time, potentially implying higher resilience; however, that coincides with a state like summer where interactions are high as well as their fluctuations due to their exponential distribution. This state has higher entropy and persistence than winter like states. Additionally, competitive species for which instability is increasing (considering both interaction and dominant eigenvalue over time) are enhancing fluctuations of cooperative species that are decreasing in abundance. Overall, an increasing divergence in the competitive-cooperative dynamics is observed over time.

- Seasonal interaction patterns are analyzed by considering the phase diagram of interaction organization. This provides an expected average behavior of interaction as a function of temperature. OIF-inferred networks for ≤10˚C and 10–15˚C groups lie on a globally stable state considering their power-law distribution of interactions underpinning persistence (or self-organized criticality that reflect functional optimality) of cross-abundance correlation (that is defining interactions) a' la [37] (see also [83]). High temperatures (20–25˚C and ≥25˚C) are locally stable because they show high value of interactions that are more frequent and regular than scale-free low temperature interactions. Ecosystem alterations can affect high temperature interactions largely and rapidly including their ability to bounce back to their dynamics. This confers a higher resilience a' la [38] but intrinsically lower global optimality considering the less organized state of interactions. Yet, higher resilience does not correspond to criticality a' la [37]. Based on the phase diagram, it is expected that with the continuous increase of ST, the fish community becomes more interactive; the ecosystem presents three different regimes, one critically stable, one unstable, and one subcritical. A critical phase transition from global stability (≤10˚C and 10–15˚C groups) to critical instability occur in between 15–20˚C and 20–25˚C. When ST continues to rise to ≥25˚C, the fish community becomes more interacting (in terms of extremes' frequency) and the ecosystem reaches a locally stable state with higher resilience but lower organization. Nodal degree is increasing from ≤10˚C to ≥25˚C from a normal to a uniform distribution associated to a non-linear transition in interaction network topology from a scale-free to a exponential/small-world network. This also emphasizes how structural network features are poorly descriptive of functional network topology underpinning ecosystem function.

- Network-based *effective salience* identified *keystone species* as the ones with the highest interaction and predictability of other species based on the defined *effective distance* inversely proportional to species interactions. Across time and seasons the most salient species are species 5 and 7 that are *Trachurus japonicus* (Horse mackerel) and *Pseudolabrus sieboldi* (Wrasse). These species (that are actually rather central in spring and fall) have the highest fitness with temperature and smaller autocorrelation over space, and yet have the highest role in the stability/instability of the ecosystem. Specifically these "Pareto" species, due to their Pareto-distributed interactions, have the largest portfolio effect, and in the studied period and community considered, they are increasing in abundance and affecting negatively cooperative species such as *Engraulis japonicus* (Japanese anchovy) whose geographical extent is much larger than competitive species. Interestingly, in winters *Rudarius ercodes* (Pigmy filefish), that is an invasive species, is the keystone species and this is likely caused by a weakly interacting community and increasing temperature creating potential niche for new species introduction. Blue/Yellow striped Goby and Chameleon Goby (sp. 11 and 12) are also quite central after Pigmy filefish (sp. 15). The latter species as well as Horse mackerel (that is reported as Near Threatened globally) and Wrasse are keystone species in the summer, which reflects the protracted species dynamics of winter and intermediate seasons.

- Dynamic temporal networks present increasing seasonality instability considering the increasing fluctuations of the dominant eigenvalue over time. During transitions seasons with high fluctuations of temperature (spring and autumn) the real part of the dominant eigenvalue from the interaction matrix is high, suggesting that the fish community is more interacting and less stable. For intermediate temperature ranges the dominant eigenvalue increases over time. This dynamical instability increases in summer and despite the dominant eigenvalue for the highest TR decreases over time the fluctuations increase. We confirm how organized interaction or degree distributions tend to stabilize food webs (captured as cross-abundance species interactions), and that average interaction strength has little

influence and information on community stability (compared also to the the effect of interaction variance and cross-correlation). Scale-free interaction networks have fewer trophic dependencies than random networks whose connectivity increases exponentially. Therefore, diversity should not be considered as a stability metric but a byproduct of interaction whose topology is fundamental for ecosystems function.

- The dominance of "weak" interactions is higher in the summer (relatively speaking in terms of distribution or probability in comparison to others than in magnitude) but extreme interactions are also present due to the persistent presence of competitive species. Higher species diversity during summer periods are associated with apparently higher dynamic stability over time and higher population fluctuations. On the contrary of [21] we emphasize how this apparently higher dynamic stability (across seasons) is decaying over time manifesting potential "critical slowing down". The divergence between dominant eigenvalue of competitive and cooperative species is also increasing over time, where the former and the latter are increasing and decreasing instability. This cross-dynamics emphasizes an overall increase in extreme competition between species group over time despite the decay in average interactions that is only related to species that are no more connected and likely "pushed out" the community. In summer, stability a' la Scheffer is higher (dominant eigenvalue is lower) but the ecosystem is less critical a la' Bak. Vice versa, in winter, higher (network) complexity is associated to lower diversity, higher persistence/optimality (a la' Bak, with a strange attractor for interactions) but lower dynamical resilience a' la Scheffer. This emphasizes the duality between optimality and temporal resilience: an ecosystem can be highly optimized but that corresponds to fragility due to the more locked configuration. Vice versa, high entropy networks recover faster from random and targeted perturbations [44]. The winter state corresponds majorly to power-law distributed interactions of competitive species (with exponentially distributed abundance) determining stabilizing portfolio (Pareto) effects. The total directed interaction is the best suited indicator for characterizing species influence on all others and largely related to species pair salience. Rising temperature fluctuations are increasing ecosystem entropy (i.e., the number of ways species can get organized without any change into the overall state determined by diversity; yet, a random network can be random in many more ways than a scale-free one) and total entropy reduction (i.e. total interactions) is decreasing, implying ecosystem's lower predictability. In conclusion, summers are more deterministic in terms of biodiversity but present much higher energy dissipation due to larger entropy/disorganization of the species interaction network. Disorganization that is increasing due to rise in temperature fluctuations and favoring competitive "black swan" species (considering their extremes and not the distribution) with higher fitness to seasonal temperature oscillations and less subjected to fishing pressure. Pareto links between these species and cooperative "white swan" species are getting weaker, yet increasing the rarity of these "weak" interactions (in magnitude rather than frequency).

- Extreme interactions are larger and more probable for competitive species, reflecting the summer dynamics where competitive are predominant and less stable (as also reflected by the dominant eigenvalue in Fig 6) on the contrary of what found by [21]. Competitive species are less stable (Bak sensu) considering both the exponentially distributed abundance and interactions—underpinning niche local determinants—as well the dominant eigenvalue (higher), but are more resilient (Scheffer sensu) precisely because of their exponential persistent dynamics (synchronized with temperature) reflected by their small-world interaction network. Cooperative species are more stable (yet more self-organized Bak sensu, where critical interactions are necessary for their punctuated equilibrium [37]) considering both the power-law abundance and interactions—underpinning neutral large-scale determinants—as

well the dominant eigenvalue (lower than competitive), but less resilient (Scheffer sensu) considering their distribution of abundance and interactions (less regular and persistent in one direction of growth) reflected in their scale-free interaction network. Temperature increase causes a disruption of Pareto links (species 5 and 7 interactions) via instability of competitive species that determines a transition in the distribution (from power-law to exponential) of cooperative species. This shift is also manifested by the temporal decrease of the relative entropy (or Kullback-Leibler divergence) or increase of the dominant eigenvalue divergence (Fig 6B) of interactions between competitive and cooperative species. Yet, average lower interactions (Figs 6 and 8A) and higher relative stability of cooperative species are not good signs, because these changes are associated to organizational decays as corroborated by the loss of criticality. These results (Fig 8) show how stability or persistence (determined by the self-organized optimality over long periods due to the likely pressure of other factors) and resilience (especially of seasonal factors such as temperature) are two different things that do not necessarily overlap. The duality between abundance and interactions exists when both are considered for different temperature groups, e.g. power-law abundance fluctuations in summer are associated to exponentially distributed interactions. Similarity or coherence between abundance and interactions exist when considering them in their whole spectrum, e.g. competitive species have an exponential distribution of abundance that is associated to exponential small-world interactions. Duality appears when considering extreme dynamics while congruence arises when considering the whole collective dynamics.

As a final chiosa, we emphasize how different ecosystem indicators and/or patterns reveal information of ecosystem change differently. Namely, a sudden collapse in interactions is reflected by the cross-seasonal interaction phase-space while a gradual change is reflected by abundance and $\alpha$-diversity patterns (such as Zipf's and Taylor's laws informing about potential community vulnerability and macrotrends). Macroecological indicators such as taxonomic diversity can just reveal trends of ecosystems but are poorly capturing any early warning of transitions. Additionally, it is important to distinguish ecosystem optimality and resilience that are complementary of each other and not necessarily overlapping as well as not coinciding with desired states. Cross-species abundance variability (via targeted eDNA and metabarcoding monitoring), informing food-web changes, is proposed as the most meaningful indicator to track ecosystem state and alterations. Causal systemic attribution of ecosystem change to global and local environmental factors should be carried out by non-linear models that consider factor and area interdependencies vs. one species/area/factor at a time approaches. This will also allow to detect critical salient species (or keystone pairs) to protect for multitrophic stability. Yet, critical interactions should guide the formulation of Multispecies Conservation Values (such as the IUCN ecosystem index [84]) and quantitative fishery policy aimed to protect marine ecosystems and improve community and species resilience to ocean warming.

## Supporting information

**S1 Table. Maizuru fish community species information.** Species ID considering the Maizuru dataset (raw data available at https://www.dropbox.com/s/q0z9hiaw9t752hx/Maizuru_dominant_sp_var.xlsx?dl=0), scientific and common name, categorization in terms of fish stock, location endemicity (native/invasive), and reported IUCN conservation status (up to December 2020). Status is provided in three categories: Not Evaluated (NE), Near Threatened (NT), Least Concern (LC). Source: http://fishbase.sinica.edu.tw/. All models, scripts for

visualizations, data and results are at https://github.com/HokudaiNexusLab/FishCommunity.
(PDF)

**S2 Table. Fish species total outgoing and incoming species interactions and mean/standard deviation of abundance.**
(PDF)

**S3 Table. Salience of top-5 species pairs for five temperature ranges and the whole period.**
(PDF)

**S1 Fig. Epdf of species abundance.** Epdf of species abundance and power-law fitting.
(PDF)

**S2 Fig. Epdf of species abundance for competitive and cooperative species.** Both are power-law with infinite mean and variance (critical regime) due to $\epsilon \leq 2$, but competitive species have a finite-size exponential decay (in the tail of the distribution) and a narrower range of criticality. This implies a more exponential set of interactions (Fig 8). Vice versa cooperative species are closer to a supercritical regime (less "heavy tail") with larger extreme abundance and higher evenness resulting in power-law interactions. This emphasizes the duality between abundance and interactions.
(PDF)

**S3 Fig. Continuous probability distribution function of species abundance.** The pdf of mean temperature across the whole period is shown as inset in the pdf of species 1.
(PDF)

**S4 Fig. Species abundance and mean temperature over time.** Abundance is show in blue while temperate in red (equal for all species). Green, yellow, an red dots are informing about increasing, stationary, and decreasing abundance with increasing temperature considering the whole period. Note that all cooperative species (species excluding 4–9), except for species 1 (i.e. *A. Aurita*), are decreasing in abundance over time, that underlines their lower fitness with temperature and yet the higher impacts due to the temperature variability in conjunction with other unexplored factors.
(PDF)

**S5 Fig. Relationships between species abundance and mean temperature.** Species abundance on log scale vs. mean temperature is linearly fitted a first degree polynomial model (red line). Yet, abundance is changing superlinearly with temperature causing nontrivial changes in species abundance.
(PDF)

**S6 Fig. Species interaction inference model intercomparison. A**: causal interaction inference CCM model developed by [63]. **B**: linear association between species computed as Pearson correlation coefficient. TE-based causal interaction inference using Kernel and Gaussian estimators (C and D, respectively) in JIDT developed by [62].
(PDF)

**S7 Fig. OIF-inferred interaction networks of top 20% greatest TEs.** The size of nodes is proportional to the Shannon Entropy of the species; the color of node is proportional to the total outgoing transfer entropies (OTE) (the higher OTE, the warmer the node's color.); the width and color of the link between species are proportional to the TE between species pairs (the higher TE, the warmer/wider the link's color/width).
(PDF)

**S8 Fig. Pdfs of nodal degree of OIF networks for the top 20% (Pareto) TE interactions. A**: Pdf of the structural degree, **B**: pdf of the in-degree, **C**: pdf of the out-degree, of species in OIF networks for the five TR groups.
(PDF)

**S9 Fig. Pdf of OTE and OTE vs. species abundance. A**: Pdf of OTE of all species for five TR groups. **B**: allometric scaling of OTE as a function of mean species abundance. The latter is the community Kleiber's law relating directed information exchange and species abundance (proportional to community metabolic rate/energy expenditure and biomass [35]). Each point refer to a species that can occur in multiple TR groups.
(PDF)

**S10 Fig. Top 5 species with the largest Shannon entropy and OTE. Left** plots show top 5 species with the highest Shannon entropy (i.e. uncertainty, or vice versa disorganized/relative information content implying lower predictability) for the five temperature ranges. **Right** plots show the top 5 most active species in terms of OTE (yet, affecting the community the most) for five temperature ranges. Due to the definition of salience (Eq 2.10) these are also largely coinciding with the species involved in the most salient pairs.
(PDF)

**S11 Fig. Allometric scaling of TE against mean and standard deviation of species abundance. A**: community Kleiber's law [35] at the species scale independent of temperature shows a non-significant trend of total directed interactions as a function of mean abundance. Vice versa, **B** shows how directed interactions are tendentially higher for rarer species whose fluctuations and mean of abundance are small. This supports the duality between abundance and interaction patterns and the fact that species popularity and species with extreme fluctuations (especially if asynchronous from the collective like for species 2) are not determining with Pareto interactions. In simple terms, species commonality does not increase inter-species interactions with all others but rather likely intra-species interactions only.
(PDF)

**S12 Fig. Time and temperature-dependent dynamical networks.** The stability of the fish ecosystem is indicated by the dominant eigenvalues of the TE interaction matrix. The total interactions are calculated as the sum of TE values in the TE matrix. Effective $\alpha$-diversity is the number of connected nodes (species) in dynamical networks derived from OIF-inferred TE matrices; statistically insignificant TEs define non-connected species. Dashed red lines emphasized the highest temperature/summer periods. **A**: Real part of the dominant eigenvalue from temporally dynamical TE matrices (blue line) and adjacency matrices (red line); **B**: Total interactions of temporally dynamical TE interaction matrices; **C**: Effective $\alpha$-diversity of temporally dynamical TE interaction matrices without threshold (certain species were taxonomically non reported at certain times); **D**: Effective $\alpha$-diversity of temporally dynamical TE interaction matrices for top 20% TEs; **E**: Real part of the dominant eigenvalue of temperature-dependent TE and adjacency matrices (blue and red line, respectively); **F**: Total interactions of temperature-dependent TE interactions; **G**: Effective $\alpha$-diversity of temperature-dependent TE interactions without threshold; **H**: Effective $\alpha$-diversity of temperature-dependent TE interactions for top 20% TEs.
(PDF)

**S13 Fig. Temporal dominant eigenvalue and functional interaction network degree for temperature groups.** The real part of the dominant eigenvalue of time-varying interaction matrices is calculated for different temperature groups. Species abundance values within

each temperature groups are used independently to calculate the dominant eigenvalue. The lowest TR has the shortest time series due to the limited duration of winters in the Maizuru Bay.
(PDF)

**S14 Fig. Continuous probability distribution functions of OTE for all species.** Interactions are inferred via the OIF model of [28] considering the whole time series.
(PDF)

**S15 Fig. Least Concern *Engraulis japonicus* (Japanese anchovy) native range.** Native species with *cooperative* behavior characterized by low directed interactions (OTE), salience (for encompassing connections), and asynchronized decreasing abundance for growing temperature. Time series are persistent, irreversible over time on average, and with low entropy. Fishing peaks (in winter) are asynchronized with temperature fluctuations but synchronized with abundance; thus, temperature is likely a second-order determinant of abundance. Abundance over time and habitat suitability (map) is power-law distributed. Source: AquaMaps; Reviewed Native Distribution Map for *Engraulis japonicus* (Japanese anchovy); Retrieved June 01, 2021 from http://www.aquamaps.org/preMap.php?cache=1&SpecID=Fis-29947 (material licensed under a Creative Commons Attribution-NonCommercial 3.0 Unported License).
(PDF)

**S16 Fig. Near Threatened *Trachurus japonicus* (Horse mackerel) native range.** Native species with *competitive* behavior characterized by high directed interactions (OTE), salience (for encompassing connections), and synchronized increasing abundance for growing temperature (implying lower ecological memory). Time series are antipersistent, reversible over time on average, and with high entropy. Fishing peaks (in summer) are synchronized with temperature fluctuations and abundance; thus, abundance fluctuations, driven by fishing and temperature, overlap and temperature is likely a first-order determinant. Abundance over time and habitat suitability (map) is more exponentially distributed with smaller autocorrelation than cooperative species. Higher connections and power-law like interactions define this species as a *keystone* species able to affect and predict ecosystem changes. Source: AquaMaps; Reviewed Native Distribution Map for *Trachurus japonicus* (Horse mackerel); Retrieved June 01, 2021 from https://www.aquamaps.org/receive.php?type_of_map=regular (material licensed under a Creative Commons Attribution-NonCommercial 3.0 Unported License).
(PDF)

**S17 Fig. Least Concern *Halichoeres tenuispinnis* (Chinese wrasse).** Invasive species, originally from SE China, with *transitory* (dynamically competitive) behavior characterized by intermediate directed interactions (OTE), salience (for encompassing connections) and mildly synchronized increasing abundance for growing temperature. Abundance over time and habitat suitability (map) is exponentially distributed with fat-tail and bimodal autocorrelation emphasizing the transitory dynamics. Source: AquaMaps; Reviewed Native Distribution Map for *Halichoeres tenuispinnis* (Chinese wrasse); Retrieved June 01, 2021 from https://www.aquamaps.org/receive.php?type_of_map=regular (material licensed under a Creative Commons Attribution-NonCommercial 3.0 Unported License).
(PDF)

**S1 File.**
(PDF)

## Author Contributions

**Conceptualization:** Jie Li, Matteo Convertino.

**Formal analysis:** Jie Li, Matteo Convertino.

**Funding acquisition:** Matteo Convertino.

**Investigation:** Jie Li, Matteo Convertino.

**Methodology:** Jie Li, Matteo Convertino.

**Project administration:** Matteo Convertino.

**Supervision:** Matteo Convertino.

**Validation:** Matteo Convertino.

**Visualization:** Jie Li, Matteo Convertino.

**Writing – original draft:** Jie Li, Matteo Convertino.

**Writing – review & editing:** Matteo Convertino.

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
