## [Decision Letter · Decision Letter 0]

3 Mar 2021

PONE-D-21-01412

Temperature-driven Organization of Fish Ecosystems and Fishery Implications

PLOS ONE

Dear Dr. Convertino,

Thank you for submitting your manuscript to PLOS ONE. After careful consideration, we feel that it has merit but does not fully meet PLOS ONE’s publication criteria as it currently stands. Therefore, we invite you to submit a revised version of the manuscript that addresses the points raised during the review process.

I tend to agree with the comments and suggestions made by both reviewers #1 and #, please address in your revision all the issues and in particular explain in more details the type of model used and their usefulness in our understanding of ecosystem processes linked to fisheries.

We look forward to receiving your revised manuscript.

Kind regards,

Andrea Belgrano, Ph.D.

Academic Editor

PLOS ONE

Journal Requirements:

"M.C. and J.L. gratefully acknowledge the funding provided by the GI-CORE Global Station

for Big-Data and Cybersecurity at Hokkaido University, Sapporo, JP. M.C. acknowledges

the FY2020 SOUSEI Support Program and Award for Young Researchers awarded by the

Executive Oce for Research Strategy to the Top 20% scientists in terms of productivity

and citations at Hokkaido University. M.C. also acknowledges the Microsoft AI for Earth

computational resources and the NIH funded Big Data to Knowledge (BD2K) 2017 Innovation

Lab Quantitative Approaches to Biomedical Data Science Challenges in our Understanding

of the Microbiome managed by the BD2K Training Coordinating Center (TCC)."

"The authors received no specific funding for this work."

Additionally, because some of your funding information pertains to [commercial funding//patents], we ask you to provide an updated Competing Interests statement, declaring all sources of commercial funding.

In your Competing Interests statement, please confirm that your commercial funding does not alter your adherence to PLOS ONE Editorial policies and criteria by including the following statement: "This does not alter our adherence to PLOS ONE policies on sharing data and materials.” as detailed online in our guide for authors  http://journals.plos.org/plosone/s/competing-interests.  If this statement is not true and your adherence to PLOS policies on sharing data and materials is altered, please explain how.

Reviewers' comments:

Reviewer's Responses to Questions

**Comments to the Author**

1. Is the manuscript technically sound, and do the data support the conclusions?

Reviewer #1: Partly

Reviewer #2: No

2. Has the statistical analysis been performed appropriately and rigorously? 

Reviewer #1: Yes

Reviewer #2: No

3. Have the authors made all data underlying the findings in their manuscript fully available?

Reviewer #1: Yes

Reviewer #2: Yes

4. Is the manuscript presented in an intelligible fashion and written in standard English?

Reviewer #1: Yes

Reviewer #2: Yes

5. Review Comments to the Author

Reviewer #1: Review on “Temperature-driven Organization of Fish Ecosystems and Fishery Implications”

This study conducted an intensive analysis of fish community dynamics driven by species interactions and sea temperature, using an information-theoretic network approach with high resolution time series data. The paper shows some important findings including the difference of community network structures and stability among different temperature ranges. I consider that most of analyses are reasonable and well sophisticated, but there are some ambiguous points to be clarified or unnatural logics and interpretations. The below comments may include my misunderstanding because I’m not so familiar with information theory or network analyses. However, more understandable explanations and clarification are necessary for publication because many readers are also not specialists in network analyses.

1. Relation of global warming to this study

In Introduction, the authors give many explanations for the impacts of ocean warming on fish and fisheries, which usually refers to long-term changes in sea temperature, whereas the analysis of this study focuses on the seasonal change of sea temperature. As there is a discrepancy of temporal scales, interpreting the results on the difference of network structure and stability among temperature zones as a scenario of temperature warming seems not to be straightforward. Although the authors also mention the effect of amplified fluctuation of sea temperature as climate change, this effect on fish network structure has not been clearly shown in this manuscript.

2. Definitions of diversity indices

The authors defined the sum of normalized abundances of each species as alpha diversity, meaning that the alpha diversity is high (low) if the abundance of each species is higher (lower) than the average. Therefore, this index is affected by not only pure species richness but also abundance. Although the evenness of abundance is an important component of biodiversity measures, I do not know any similar biodiversity index based on the relative abundance of each species. I roughly checked the cited article (Li and Convertino 2019), but this previous study seemed to use the probability to find one species instead of the relative abundance of the species. So, I suppose quite different indices between this and previous studies. Since this study selected 15 species, this index is likely to reflect species abundance rather than species richness in my speculation. This may make the result on the relationship between temperature and alpha diversity (Figure 1C) misleading.

The authors also used the Shannon entropy index based on normalized abundance of each species (line 310) and explained that the sum of this per-species index is proportional to the Shannon community index (lines 312-316). This is not trivial, and a proof is needed, however. I think that the Shannon community index is usually based on the proportion of a species within a community rather than the probability of which a species is observed in the community. Rationale of using these two measures (alpha diversity and Shannon index) with reference to previous studies is necessary.

3. Separation of regimes

The result on two regimes in the fish network for 15-20℃ (Fig. 5) is interesting, but somewhat seems arbitrary. Why were probability distribution functions mixed only for 15-20℃? Any supportive evidence or criterion of multiple regimes?

4. Degree of network scales and fish community stability

I cannot understand the argument that the less scale-free network observed in 15-20℃ leads to metastable states in contrast to the more scale-free network in lower temperature ranges. If species is more evenly distributed in a less scale-free network than a more scale-free network, it might be possible that the less scale-free community network would have greater stability.

5. Relationship between local stability and temperature

Although the study used the number and magnitude of real part of dominant eigenvalue to judge the dependence of local stability on temperature zones (Fig. 6), it is not fair because the number of points is different among the temperature zones. Although the highest temperature zone certainly has the largest real part of the dominant eigenvalue, it had at least 15 red points, and the proportion of unstable points is 2/15. By contrast, the lowest temperature zone appears to have only three blue points, resulting 1/3 unstable points. It is therefore difficult to conclude that increasing temperature makes the fish community unstable from this result. A fairer comparison is necessary.

6. Construction of texts

There are several texts that are not well constructed and redundant in this manuscript. In Introduction 1.1, the first to third paragraphs all give too long explanations for the impacts of ocean warming on fish and fisheries with many references but without an expansion of story. A reference (Free et al. 2019) is explained twice in the first and second paragraphs, which is not usual. This writing style obscures potential problems of purely biological models in previous studies and the advantage of optimal information flow model used in this study. Please try to more clarity the disadvantage or limitation of previous approaches (purely biological models) and the merit of information-theoretic network approach by reconstructing Introduction.

Introduction 1.3, which explains about the stability, sustainability and management of fish ecosystems, is related to a general problem around the globe rather than a specific issue to be addressed in this study. I suggest including this section in a more upper part of Introduction rather than the last part.

Moreover, the authors tend to include the explanations in Results that should be included in Material and Methods or Figure legends (e.g., lines 504-511, line 626-633). See also the below minor comments.

Minor comments

Line 83: Usually, only the last name is used (Cheng et al. (2013) found…)

Line 117: macroscopic analyses = purely biological model?

Line 124: Does the mutual relationship assume the effect of fish species on sea temperature as well? If so, why?

Line 136: The fish community is managed? Or surveyed by Kyoto University?

Lines 227-228: Why can the authors conclude that ignoring rare species has little impact?

Lines 238-242: A more detailed description is necessary regarding how to deal with time series data categorized into five temperature ranges. For example, the time series from April to May and that from May to October are equally treated or different?

Line 283 and Table S1: How do the authors distinguished species into fish stocks? Why is Aurelia sp. in the fish stock group?

Line 284: Does invasive species mean non-native species in this context?

Line 310: What does the event indicate?

Lines 348-350: A clearer explanation is preferable. TE from Kernel model presents ‘more’ similar patten to that of CCM and Pearson correlation ‘than TE from gaussian model’, is this understanding right? Is this because the gaussian model is likely to be affected by outliers in data? I’m also concerned about why the similar result to CCM and Pearson can be a reason to choose the Kernel model? Isn’t any model selection approach applicable?

Figure 2: Why does E. japonicus have a higher abundance than EP in lower temperature ranges?

Line 465 and Figure 2: E. japonicus appears to have a convex upward relationship against temperature with a peak around 15℃ rather than a linear relationship.

Lines 488-490 and Figure 3: Is the difference of scaling exponents between temperature ranges statistically significant? It seems quite trivial.

Lines 490-494: I suppose that the scaling exponents and sensitivity to temperature are not necessarily linked.

Lines 505-511: Able to remove this explanation because it has already been included in the figure legend.

Lines 625-630: The explanation about dominant eigenvalues and stability should be included in Material and Methods rather than Results.

Figure 5: No ‘F’ in the upper panel.

Line 780: These two references are both on terrestrial systems, but not the ocean.

Lines 786-787: From which analysis the authors can conclude increasing temperature fluctuation leads to biodiversity loss?

Lines 839-841: Is there any possibility that species 7 makes ecosystem unstable rather than stable?

Lines 865-870: In my understanding, dominant eigenvalues provide information only on local stability, but not stability of multiple states.

Reviewer #2: This study analyzed the timeseries data of marine fish community, which was published in an earlier paper, and looked at how community structure changes with temperature. The study identified some structural and dynamical changes using TE-based (causality) network analysis. The main text needs extensive re-editing, as many parts that are located in Results actually interpret, or speculate based on, the result and should be moved to Discussion. There are also some major flaws in interpretating the result, including the main discussion that relate network structural changes to temperature. Plus, the ecological relevance of TE-based stability or interactions was not clear, which requires more effort to make clearer the major finding of present study and difference from earlier similar studies (such as Ushio et al. 2018).

Major Points

#1 (“information dynamics of ecosystems”, li. 116) Ecologists, possibly the main readers of this article, might not be used to the term, information dynamics. Please rephrase it to help their ecological interpretations.

#2 (“optimal information flow model”, li. 135) What is OIF model? It should be explained for non-specialist readers. To my understanding, it is relevant, but does not directly represents interspecific flows of energy or materials, which most ecologists are interested in. It would be helpful if the authors explain what “directed flow of information” means in the context of ecology. Probably, the authors might relate it with “causality” for better understanding by general readers. An explanation that might help readers is in Discussion (“predictive interaction networks”, li. 849).

#3 It would be essential for general readers to capture the relevance of “information flow” to any ecological processes. The author would explain what it would ecologically mean that Sp. A has a “larger” information flow to Sp. B than to Sp. C. Can we interpret it as Sp. A density has larger per-capita impact on Sp. B than on Sp. B? Does it mean that energy/material flow from Sp. A to Sp. B is larger than flow to Sp. C? I guess not.

#4 The authors may also explain what is expected in their analysis between two species A and B when they are affected by a common environmental factor C (which might not be observed).

#5 (li. 155, li. 232-242) I understood that the community networks were reproduced for the different TR groups and that some structural differences were identified between them. However, how can the authors conclude that those network structural changes were “caused” by temperature changes? It only tells that there might be a seasonal difference. To conclude that it is due to different temperature, the authors should conduct an additional analysis to detect the information flow from temperature to the network structure, I guess.

#6 TE based methodology would detect an interaction from Sp. A to Sp. B when the former species indirectly affects the latter via another Species C that mediate the two and the detection probability would be higher when the individual links are stronger. Given this, I would speculate that when strengths of interactions are stronger, the method would detect more interactions from the same system, even if the number of link is in reality not changed. Is it right? Is there possibility that the pattern you found is explained by this?

#7 Ushio et al. (2018) is basically cited only as their data resource, However, the central topic of their study (“to investigate ecosystem stability by inferring species interactions and reconstructing networks”, li. 202-203) is actually the very topic Ushio et al. (2018) dealt with. I think the authors should refer the paper in a more correct way.

#8 (“Ignoring rare species does not significantly change the marine fish ecosystem and results of this work”, li. 227-228) I wondered why the authors can conclude without analysing the data with more species included.

#9 (“native and invasive species groups”, li. 283-285) The authors categorized Plotosus lineatus, Siganus fuscescens and Rudarius ercodes as invasive species with no statements about how it was judged. The author should make clear based on which data or references they were categorized. The same is true for “Fish Stock” categorization.

li. 378) The author should explain what is a rationale of utilizing eigenvalues of TE matrix (not Jacobian matrix based on dynamics of biomass or population densities) as an indicator of system’s stability. None of the studies referred in the MS (Coyte et al., 2015; Ushio et al., 2018; Stone, 2018) do not use TE matrix to evaluate community stability. How relevant is TE-based stability to population dynamical stability?

#10 (“This result means that increasing fluctuations of sea temperature lead to biodiversity loss for the fish community”, li. 451-452) There might be a correlation, but no reason to believe there is a causation between them. Such a speculation should be removed from Result section.

#11 (“The highest total abundance of species in EP, FS and native groups in 2009 is likely owing to the least temperature fluctuation in this year”, li. 457-458) Again, any sentences with speculation and interpretation in Result should be moved to Discussion. The same applies to many other parts (for example, “This finding means…” (li. 536-540)) and the main text seems to need extensive re-constructions.

#12 (“Eigenvalues and eigenvectors…(Woolf, 2009)”, li. 625-633) This part explains the methodology and should be moved to Materials & Methods.

Minor Points

#13 (“seafood web”, li. 105) Probably “sea food web”?

#14 (“Purely biological analyses”, li. 125) What does it mean? Ecology is not biology here?

#15 (li. 328) Not “Schreiber (2000)” , but “(Schreiber 2000)” here.

#16 (“we exploit eigenvalues of TE matrix as an indicator to evaluate network stability”,

#17 (“Pseudolabrus.sieboldi”, li. 838) Remove “.” Between “Pseudolabrus” and “sieboldi”.

6. PLOS authors have the option to publish the peer review history of their article (what does this mean?). If published, this will include your full peer review and any attached files.

Reviewer #1: No

Reviewer #2: No

---

## [Author Response · Author response to Decision Letter 0]

7 Sep 2021

Reviewer #1: Review on “Temperature-driven Organization of Fish Ecosystems and Fishery Implications”

This study conducted an intensive analysis of fish community dynamics driven by species interactions and sea temperature, using an information-theoretic network approach with high resolution time series data. The paper shows some important findings including the difference of community network structures and stability among different temperature ranges. I consider that most of analyses are reasonable and well sophisticated, but there are some ambiguous points to be clarified or unnatural logics and interpretations. The below comments may include my misunderstanding because I’m not so familiar with information theory or network analyses. However, more understandable explanations and clarification are necessary for publication because many readers are also not specialists in network analyses.

1. Relation of global warming to this study

In Introduction, the authors give many explanations for the impacts of ocean warming on fish and fisheries, which usually refers to long-term changes in sea temperature, whereas the analysis of this study focuses on the seasonal change of sea temperature. As there is a discrepancy of temporal scales, interpreting the results on the difference of network structure and stability among temperature zones as a scenario of temperature warming seems not to be straightforward. Although the authors also mention the effect of amplified fluctuation of sea temperature as climate change, this effect on fish network structure has not been clearly shown in this manuscript.

Thank you very much for these comments. We fully understand the reviewer’s concern about the discrepancy of temporal scales. In this study, we considered both short-term fluctuations of sea temperature and long-term ocean warming. For short-term fluctuations, the time series were partitioned into five temperature ranges (TR) to study particular features of TR groups. For long-term ocean warming, the whole time series were considered to analyze the global variances of species diversity and populations, and to investigate temporally and temperature-dependently dynamical networks in consideration of long-term changes of sea temperature. The accumulation of small anomalous fluctuations within short periods of time caused by climate change results in long-term ocean warming, whereas most studies discussed the effects of climate change on ecosystems only considering the global long-term scale. 

As for this point, we reorganized this section of the introduction. We first talked about the effect of long-term ocean warming on marine fish ecosystems, then included literature reviews on studying the effects of long-term and short-term seasonal fluctuations on marine fish ecosystems. We stated that the study of fish ecosystems only considering species abundance and macroecological indicators is far from being sufficient to deeply understand the feedbacks of the fish community in response to the change of sea temperature (taken as the potential leading cause or indicator of change). Therefore, we propose a novel analysis based on dynamical networks that look into the topology and stability of species interactions. 

2. Definitions of diversity indices

The authors defined the sum of normalized abundances of each species as alpha diversity, meaning that the alpha diversity is high (low) if the abundance of each species is higher (lower) than the average. Therefore, this index is affected by not only pure species richness but also abundance. Although the evenness of abundance is an important component of biodiversity measures, I do not know any similar biodiversity index based on the relative abundance of each species. I roughly checked the cited article (Li and Convertino 2019), but this previous study seemed to use the probability to find one species instead of the relative abundance of the species. So, I suppose quite different indices between this and previous studies. Since this study selected 15 species, this index is likely to reflect species abundance rather than species richness in my speculation. This may make the result on the relationship between temperature and alpha diversity (Figure 1C) misleading.

Thank you for this comment. We respectfully disagree with the reviewer’s interpretation about the definition of alpha diversity indicated as equation 2.3. We understand that alpha diversity is a local measure referring to the average species diversity in a habitat or particular area, that is independent on species abundance. Equation 2.3 mathematically formulates this definition. If species i appears at time t, (species abundance xi(t) is non-zero, xi(t)0 is 1), species i will be counted in alpha diversity at time t, otherwise xi(t)0 is 0 which means that species i will not be counted at time t. Therefore, this definition (equation 2.3) is equivalent to the one we defined in our previous article (Li and Convertino 2019).

The authors also used the Shannon entropy index based on normalized abundance of each species (line 310) and explained that the sum of this per-species index is proportional to the Shannon community index (lines 312-316). This is not trivial, and a proof is needed, however. I think that the Shannon community index is usually based on the proportion of a species within a community rather than the probability of which a species is observed in the community. Rationale of using these two measures (alpha diversity and Shannon index) with reference to previous studies is necessary.

Thank you for this valuable comment. We fully understand the reviewer’s concern about the relationship between the sum of Shannon entropy of species abundance and Shannon community index. We clarify that the sum of the Shannon entropy index is unnecessarily proportional to the Shannon community index because these two concepts consider two independent variables (species abundance and richness). Species richness does not take into account species abundances or their relative abundance distributions. Shannon entropy of species abundance in this paper is used to quantify the size of nodes in networks. Since we didn’t talk about the sum of Shannon entropy and Shannon diversity index in this manuscript finally, we deleted the description about this concept in that paragraph.

We modified section 2.4.1 as follows:

“In this study, we calculate Shannon entropy for each species in the fish community based on pdf estimation of species abundance. Shannon entropy of species $i$ is defined as $H(X_i)=-\\sum_{m=1}^{v}p(x_i(m))log_{2}{p(x_i(m))}$, where p(xi(m)) is the probability of an event $m$ of unique normalized abundance of species $i$, $v$ is the number of all unique events of abundance. Note that event is a concept in probability theory, here we considered a unique number of species abundance observed in the time-series data as an event. The species-specific Shannon entropy implies the uncertainty of species observations (samples) over time, and is used as one of information-theoretic variables to define the structure of inferred networks (the size of nodes).”

3. Separation of regimes

The result on two regimes in the fish network for 15-20℃ (Fig. 5) is interesting, but somewhat seems arbitrary. Why were probability distribution functions mixed only for 15-20℃? Any supportive evidence or criterion of multiple regimes?

Thank you very much for this interesting comment. We fully agree with the reviewer’s concern. The paper was unclear about the identification of the break point cutting the EPDF and the meaning of EPDFs in a network sense. Each probability distribution function of species interactions (and related EPDF) is associated to a distinct stochastic dynamics or regime that may be more or less stable (e.g. an exponential and a power-law pdf is associated to a Poisson and a scale-free network that are less and more stable, respectively); hybrid pdfs exist and we analyzed them accordingly. 

In this paper, ymin in equation 2.1 is always used as the break point Ybreak in equation 2.2 to isolate different regimes whose scaling exponents characterize different patterns of probabilistic distribution. ymin is an estimated lower bound for which the power-law holds. We compared the ymin yielded from the EPDF estimation model to median value of TE values. ymin -s for 15-20℃ (0.584) and 20-25℃ (0.653) are much higher than the median values of TE (0.263 and 0.205), while those for other TR groups are close. It means that most values in TE matrix (82% and 88%) for these two groups are not captured by the fitting model. Therefore, we divided the EPDF into two regimes for 15-20℃ and 20-25℃ groups. However, the power-law fitting model gives very close scaling exponents for the two divided regimes in the 20-25℃ group. Thus, only 15-20℃ group has two evident regimes with very distinct scaling exponents showing obviously different distribution patterns.

As for this point, we included the following content in section 2.4.2: 

“In this study, a break point is set to divide the EPDF of species interactions for 15-20℃ group into two pieces since the break point Ybreak (ymin) estimated from the model is much higher than the median value of species interactions, yet the divided two regimes have quite different scaling exponents (probabilistic patterns).”

and the following content in section 3.3: 

“The distribution of species interactions for 15-20_C group is divided into two sections by a break point (ymin=0.584) estimated from power-law fitting model since the estimated ymin is much higher than the median value (0.263) of TE, yet the scaling exponents of two divided EPDF sections are very much different (-0.357 and -4.468, respectively) (Johannesson et al., 2006; James et al., 2018).”

4. Degree of network scales and fish community stability

I cannot understand the argument that the less scale-free network observed in 15-20℃ leads to metastable states in contrast to the more scale-free network in lower temperature ranges. If species is more evenly distributed in a less scale-free network than a more scale-free network, it might be possible that the less scale-free community network would have greater stability.

Thanks a lot for this comment. We fully understand the reviewer’s concern. In this paper, we recognized the properties of networks considering the distribution patterns of species interactions. OIF-inferred species interactions determine the structure of interaction networks that strongly correlates with the stability of ecosystems. 

On the one hand, a more scale-free network is determined by how well the power-law model fits the EPDF of OIF-inferred species interactions. This power-law feature describes a phenomenon that there are some most interacting hubs (clusters) in the network that are closely followed by a large number of nodes (species) with weak interactions. This hierarchy allows for a fault tolerant behavior. Skewness toward weak interactions enhances stability (van Altena et al. 2016), and the scale-free network shows a higher resilience and robustness to a perturbation or stress.

On the other hand, evenness of species distribution does not necessarily imply stability. Even distribution seems to make contributions to species populations, richness and evenness, while brings fiercer interspecific and intraspecific competition and increases the uncertainty (complexity) of the species community with respect to information-theoretic understanding. Actually, the stability-diversity-complexity debate has persisted as a central focus of theoretical ecology for half a century. Before 1970s, the predominant view was that ecological complexity could stabilize ecosystems. Another theory put forth by Robert May after 1970s states that complex random systems are less likely to recover from small perturbations than simpler ones. This longstanding debate indicates that the stability of ecosystems doesn’t only correlate with several indicators, but all relevant ingredients to the ecosystems. Thus, we applied the information-theoretic OIF model to study the ecosystem considering both macroecological indictors, species interactions and environmental factors (sea water temperature).

Landi, P., Minoarivelo, H.O., Brännström, Å. et al. Complexity and stability of ecological networks: a review of the theory. Popul Ecol 60, 319–345 (2018).

van Altena, Cassandra, Lia Hemerik, and Peter C. de Ruiter. "Food web stability and weighted connectance: the complexity-stability debate revisited." Theoretical Ecology 9.1 (2016): 49-58.

Jacquet, Claire, et al. "No complexity–stability relationship in empirical ecosystems." Nature communications 7.1 (2016): 1-8.

5. Relationship between local stability and temperature

Although the study used the number and magnitude of real part of dominant eigenvalue to judge the dependence of local stability on temperature zones (Fig. 6), it is not fair because the number of points is different among the temperature zones. Although the highest temperature zone certainly has the largest real part of the dominant eigenvalue, it had at least 15 red points, and the proportion of unstable points is 2/15. By contrast, the lowest temperature zone appears to have only three blue points, resulting 1/3 unstable points. It is therefore difficult to conclude that increasing temperature makes the fish community unstable from this result. A fairer comparison is necessary.

Thank you very much for this comment. The manuscript was very unclear about the relationship between system stability and sea temperature considering the distribution of eigenvalues (Fig. 6). To better distinguish the scatters overlapped in the complex plane, we replaced the solid scatters with transparent ones. We also rewrote the following paragraph to analyze the ecosystem stability and evolution in terms of the distribution of eigenvalues:

“Eigenvalues of species interaction matrix for each TR group are scattered together in a complex plane (Fig. 6). This method displays the position of all eigenvalues in the form of an ellipse (Coyte et al. 2015; Stone, 2018). There are 8 eigenvalues for ≤10℃ group, 14 eigenvalues for 10-15℃ group, 14 eigenvalues for 15-20℃ group, 15 eigenvalues for 20-25℃ group and 15 eigenvalues for ≥25℃ group. Eigenvalues correlate with the stability and resilience of the fish ecosystem in both magnitude and variance. When only considering the eigenvalues with positive real parts that may result in instability, there are 2 eigenvalues with positive real parts (2.028 and 0.009) for ≤10℃ group, 5 eigenvalues with positive real parts (2.262, 0.002, 0.002, 0.003 and 0.0003) for 10-15℃ group, 1 eigenvalue with a positive real part (4.998) for 15-20℃ group, 1 eigenvalue with a positive real part (4.934) for 20-25℃ group and 2 eigenvalues with positive real parts (6.143 and 0.052) for ≥25℃ group. It is observed that each TR group has eigenvalues with positive real parts that may bring instability to the ecosystem, while positive real parts are greater for higher TR groups (especially for 15-20℃,20-25℃ and ≥25℃ groups). Sea temperature is quite likely the strong causal factor. This result confirms the finding that the fish ecosystem presents more instability for 15-20℃, 20-25℃ and ≥25℃ TR groups.” 

6. Construction of texts

There are several texts that are not well constructed and redundant in this manuscript. In Introduction 1.1, the first to third paragraphs all give too long explanations for the impacts of ocean warming on fish and fisheries with many references but without an expansion of story. A reference (Free et al. 2019) is explained twice in the first and second paragraphs, which is not usual. This writing style obscures potential problems of purely biological models in previous studies and the advantage of optimal information flow model used in this study. Please try to more clarity the disadvantage or limitation of previous approaches (purely biological models) and the merit of information-theoretic network approach by reconstructing Introduction.

Introduction 1.3, which explains about the stability, sustainability and management of fish ecosystems, is related to a general problem around the globe rather than a specific issue to be addressed in this study. I suggest including this section in a more upper part of Introduction rather than the last part.

Moreover, the authors tend to include the explanations in Results that should be included in Material and Methods or Figure legends (e.g., lines 504-511, line 626-633).

Thanks a lot for these comments. We totally agree with the reviewer’s concern. We reorganized the section of introduction. Firstly, published statistics are listed in the first paragraph to show the fact that ocean warming caused by climate change has been threatening the sustainability of marine ecosystems, and emphasize the importance of studying the effects of the fluctuations of sea temperature on marine ecosystems, especially on fish communities and fisheries. Currently, most studies on these topics are performed only by analyzing time-series data at global long-term and seasonal short-term temporal scales. Results from these studies can individually exhibit linear or non-linear relationships between species diversity, abundance and the change of environmental factors (for instance, sea temperature), but are quite difficult to explicate how the change of sea temperature affects the fish community, and how the fish community responds to the abnormal fluctuations of sea temperature from the perspective of holistic system and its collective behavior. 

Secondly, we review the stability, sustainability and management of marine fish ecosystems, and highlight the importance of investigating the fish community interactions by considering them as an integral ecosystem component and analyzing the collective behavior at multiple temporal scales. 

Thirdly, in addition to ecological analyses, we suggest performing this work via burgeoning complex network models and data science. The fundamental work of modeling ecosystems using complex networks is to detect causal interactions between species. In reality, it is extremely hard to infer true causality in real-world ecosystems due to the intrinsic lack of knowledge about the “true” reality of a system especially for highly complex non-linear systems driven by complex environmental forcing. To solve this problem, we introduce an optimal information flow model that considers causality as information flow and predictability, making the causality practical to quantify mathematically. 

We agree with the reviewer’s comment on the introduction 1.3. The explanations about the stability, sustainability and management of fish ecosystems were placed in introduction 1.2 to stress the importance of studying the fish ecosystem in a system view. Then we introduce our developed information-theoretic network model after introduction 1.2.

We fully agree with the reviewer’s comment about some explanations in Results. We deleted line 505-511 in the main manuscript and improved this description in the caption of Figure 4. We also moved line 625-633 to the first paragraph of section 2.4.3.

See also the below minor comments.

Minor comments

Line 83: Usually, only the last name is used (Cheng et al. (2013) found…)

Yes, we corrected these citations with wrong format. Thank you for noticing it.

Line 117: macroscopic analyses = purely biological model?

Thank you for this comment. In this manuscript, macroscopic analysis means an analytical procedure that treats the fish community as a whole ecosystem without considering the specific individual species in the fish community. We wanted to use “Purely biological model” to describe a method to study the fish ecosystem only considering species abundance, composition and richness, in an ecological way. We deleted the expression of “purely biological model” since this is unclear and misleading. Actually, in this study, we studied individual populations species by species (species level), macroecological indicators of α diversity (community level), and species or community interactions over time (ecosystem level), simultaneously. 

Line 124: Does the mutual relationship assume the effect of fish species on sea temperature as well? If so, why?

Thanks for this interesting comment. Ecologically speaking, even though fish populations do not directly affect sea temperature, we can find some clues about sea temperature in fish populations. These clues can be interpreted as the information of sea temperature encoded in fish populations over abundance time records. Therefore, observations of fish populations can be used to inversely predict the change of sea temperature. 

Line 136: The fish community is managed? Or surveyed by Kyoto University?

Thank you for this comment. These long-term data sets were collected at Maizuru Bay, Japan. This coastal area is surveyed by Kyoto University. There is a research station affiliated by Field Science Education and Research Center, Kyoto University. It aims to conduct multidisciplinary studies on coastal organisms and their environment.

Lines 227-228: Why can the authors conclude that ignoring rare species has little impact?

Thanks for this comment. In short, predictions of other species interactions would not be affected by rare species because these contain very little information in common species, as already found by the study below. Rare species would provide small interactions and yet have little effects on the pdf of interactions and network topology, provided they are not strongly interacting species such as predatory invasive species that sporadically enter the system. Certainly, there is no model that can account for these sporadic events unless they are artificially introduced in the analysis, but that would introduce a set of assumptions that cannot be verified. The sampling effort of the Maizuru Bay is highly reliable and published in top-profile journals.

We clarified this aspect in the second paragraph of section 2.1 as below: 

“since abundance dynamics of rare species contained within other species is typically very small, the inclusion of rare species does not alter the main ecosystem interaction topology beyond just adding poorly interacting species as found in Ushio et al, (2018) and Li and Convertino (2021).” 

Ushio, M., Hsieh, Ch., Masuda, R. et al. Fluctuating interaction network and time-varying stability of a natural fish community. Nature 554, 360–363 (2018).

Li, J., Convertino, M. Inferring ecosystem networks as information flows. Sci Rep 11, 7094 (2021).

Lines 238-242: A more detailed description is necessary regarding how to deal with time series data categorized into five temperature ranges. For example, the time series from April to May and that from May to October are equally treated or different?

Thanks a lot for this comment. We did not explain the data categorization considering five temperature ranges clearly. Then, in the manuscript, we included the following sentence in Section 2.1 (Paragraph 3):

“The data categorization picks out the abundance observations at time points when the mean sea temperature is within a specific temperature range and rearranges the selected abundance observations in time order, leading to a new time series corresponding to the temperature range.”

Since the data categorization picks out abundance observations only considering the mean temperature of the time point (without considering the time itself), time series from April to May and those from May to October are equally treated.

Line 283 and Table S1: How do the authors distinguished species into fish stocks? Why is Aurelia sp. in the fish stock group?

Thanks for this comment. We categorized these species into fish stocks according to the website of fishbase that is a global fish species database. Take Aurelia aurita as an example, please refer to: 

https://www.sealifebase.ca/summary/Aurelia-aurita.html

we can find that this species is considered commercial since it has “human uses”. As for this point, we included the link of the “fishbase” website in the paper (Section 2.3).

Line 284: Does invasive species mean non-native species in this context?

Yes, that is correct. Thanks.

Line 310: What does the event indicate?

Thank you for this comment. Event is a concept in probability theory. A specific number of species abundance observed at a time point is defined as an event. As for this point, we added this sentence in the section 2.4.1.

“Note that event is a concept in probability theory, here we considered a unique number of species abundance observed in the time-series data as an event.”

Lines 348-350: A clearer explanation is preferable. TE from Kernel model presents ‘more’ similar patten to that of CCM and Pearson correlation ‘than TE from gaussian model’, is this understanding right? Is this because the gaussian model is likely to be affected by outliers in data? I’m also concerned about why the similar result to CCM and Pearson can be a reason to choose the Kernel model? Isn’t any model selection approach applicable?

Thank you for this valuable comment. We fully agree with the reviewer’s understanding. Gaussian model provided a quite different pattern than other 3 models. It is possibly because Gaussian model is subject to linear-model assumption, assuming linear interactions between variables. Outliers in data may improve the non-linearity of observations, leading to the decrease in performance of Gaussian model. Pearson correlation coefficient is a symmetrical measure of linear correlation between two sets of data. This measure only captures linear relationship between variables, it is not capable of distinguishing directed interactions, either. ρ from CCM and TE from Kernel model present similar patterns, while the OIF-inferred heatmap presents larger gradient of interactions that highlight the divergence in fish populations of species 4-9 from other species compared to the heat map from CCM model. As for these points, we modified this paragraph as below:

“In Fig. S5, Gaussian model provides a quite different pattern than other three models since Gaussian model is subject to linear-model assumption, assuming linear interactions between variables (Lizier,420 2014). Pearson correlation coefficient is a symmetrical measure of linear correlation between two sets of data. This measure only captures linear relationship between variables, it is not capable of distinguishing directed interactions, either. Even though ρ from CCM and TE from OIF with Kernel model present similar patterns, the OIF-inferred heat map from Kernel presents larger gradient of interactions that highlight the divergence in fish populations of species 4-9 from other species compared to the heat map from CCM. This divergence in the distribution of species populations can be observed in Fig.S2. It is worth noting that species 4-9 are all native species (see Table S1). Therefore, OIF with Kernel estimator allows a better identification of species clusters considering gradients of inferred interactions. Additionally, TE from Kernel model is able to estimate some weak observed interactions such as of species 2 with others, while CCM essentially consider null interactions for these species. Therefore, Kernel model is selected as the TE estimator in the OIF model (Li and Convertino, 2021).”

Figure 2: Why does E. japonicus have a higher abundance than EP in lower temperature ranges?

Thanks for this comment. EP is defined as the total abundance of all species in the fish community that should be higher than FS, Aurelia and E. japonicus. We checked the data and Figure 2 and found that the data calculation was correct. However, the value of the first point on the left corresponding to the lowest temperature for EP (153) is very close to that for E. japonicus (150). The yellow point is almost covered by the red point since the red one is at the first front.

Line 465 and Figure 2: E. japonicus appears to have a convex upward relationship against temperature with a peak around 15℃ rather than a linear relationship.

We greatly appreciate the further insight of the reviewer and fully agree with the comment. As for this point, we modified the sentence as below:

“Therein, the abundance of species 2 (engraulis japonicus), as an exception, increases with the rise of sea temperature and reaches a peak around 15℃, then drops as the sea temperature continues to rise.” 

Lines 488-490 and Figure 3: Is the difference of scaling exponents between temperature ranges statistically significant? It seems quite trivial.

Lines 490-494: I suppose that the scaling exponents and sensitivity to temperature are not necessarily linked.

Thank you very much for these two comments.

Figure 3B showing Taylor’s power law describes the relationship between the variance of species abundances and their mean abundances. The regression of the standard deviation vs. mean abundance on a log-log scale presents slopes of less than 1. This pattern provides an understanding on general ecological processes affecting many species related to many ecological problems involving the stochastic fluctuations of species population densities, habitat changes (Kilpatrick and Ives, 2003). In this paper, we modified the paragraph and added a citation as below:

“The regression of log standard deviation vs. log mean abundance for all TR groups gives lines with positive slopes less than 1. Positive slopes address that the total variability of species abundance increases with the rise of mean species abundance. Slopes less than 1 suggest that the per capita variability of species abundance for all TR groups decreases with the increasing mean species abundance (Kilpatrick and Ives, 2003). Therein, slopes of scaling law for lower TR groups (≤10℃, 10-15℃, 15-20℃) slightly increase with the increasing sea temperature, and are obviously higher than those for higher TR groups (20-25℃, ≥25℃). It indicates that the variability of species abundance is on average higher for lower TR groups compared to higher TR groups. This result confirms the finding from Fig. 1C that species abundance in lower TR groups (≤10℃, 10-15℃, 15-20℃) is more sensitive to the change of sea temperature compared to higher TR groups (20-25℃, ≥25℃). The sharp decrease of scaling exponents implies that the fish community experiences a significant change in species populations and collective behavior around 20℃.”

Kilpatrick, A. M., and A. R. Ives. "Species interactions can explain Taylor's power law for ecological time series." Nature 422.6927 (2003): 65-68.

Lines 505-511: Able to remove this explanation because it has already been included in the figure legend.

Yes, we agree with the reviewer. We removed this explanation in the main manuscript, and improved the caption of Figure 4.

“The size of node is proportional to the Shannon Entropy of species abundance, the color of node scales with the total outgoing transfer entropy (OTE) of species (the greater the OTE of a species is, the warmer the color of the node.); the width and color of the link between species are proportional to species interactions computed as TE (the greater the TE is, the warmer (wider) the link's color (width) is.). The arrow of links stands for the direction of species interaction (TE).”

Lines 625-630: The explanation about dominant eigenvalues and stability should be included in Material and Methods rather than Results.

Yes, we fully agree with the reviewer’s comment. We moved this part to the first paragraph of section 2.4.3. Thanks a lot for noticing it.

Figure 5: No ‘F’ in the upper panel.

Yes, we added it. Thanks for noticing it.

Line 780: These two references are both on terrestrial systems, but not the ocean.

Thanks a lot for this comment. We fully agree with the reviewer and understand this problem. We placed some literatures relevant to marine ecosystems there.

Lines 786-787: From which analysis the authors can conclude increasing temperature fluctuation leads to biodiversity loss?

Thanks for this comment. We totally agree with the reviewer. Even though, as one of the reasons for the biodiversity loss, there might be a correlation between biodiversity loss and increasing temperature fluctuations, we can’t directly make the conclusion. Therefore, we modified the relevant contents in sections of Results, Discussion and Conclusions, as follows:

“This result implies that the change of biodiversity loss in the fish community may relate to the increasing fluctuations of sea temperature.” 

“These analyses capture the seasonal fluctuations of sea temperature, species populations and diversity, and the results support a speculation that the change of biodiversity in the fish community can be a biological response to the increasing fluctuations of sea temperature.”

“Accordingly, the biodiversity loss in the fish community may relate to the increasing fluctuations of sea temperature.”

Lines 839-841: Is there any possibility that species 7 makes ecosystem unstable rather than stable?

Thanks for this comment. Yes, there is. We intuitively suppose that species 7 plays a positive role in making the ecosystem stable since this species is a native species and has lived in Northwest Pacific area for a long time. Please refer to:

(https://www.fishbase.se/Country/CountrySpeciesSummary.php?c_code=392&id=57440)

Lines 865-870: In my understanding, dominant eigenvalues provide information only on local stability, but not stability of multiple states.

Thanks a lot for this comment. We fully agree with the reviewer and understand the reviewer’s concern. “Metastability” was misleading here. Dominant eigenvalue provides information only on local instability which is a specific equilibrium point the system is around and will return to on its own in some time after perturbations. We modified this part and corrected a typo as follows:

“Dominant eigenvalue provides information on local stability which is a specific equilibrium point the system is around after perturbations and will return to on its own in some time. Its fluctuations over time reveal less stability for the fish ecosystem during the period with higher fluctuations of sea temperature in one year. It means that given a specific stable state (equilibrium point), the ecosystem within this period is easier to oscillate around the equilibrium point and may take more time to recover. This result is in line with the relatively unstable state of 15-20℃ and 20-25℃ groups concluded in section 3.3, since these two temperature ranges are more likely to appear in spring and autumn which have relatively high temperature fluctuations in one year.”

 

Reviewer #2: This study analyzed the timeseries data of marine fish community, which was published in an earlier paper, and looked at how community structure changes with temperature. The study identified some structural and dynamical changes using TE-based (causality) network analysis. The main text needs extensive re-editing, as many parts that are located in Results actually interpret, or speculate based on, the result and should be moved to Discussion. There are also some major flaws in interpretating the result, including the main discussion that relate network structural changes to temperature. Plus, the ecological relevance of TE-based stability or interactions was not clear, which requires more effort to make clearer the major finding of present study and difference from earlier similar studies (such as Ushio et al. 2018).

Major Points

#1 (“information dynamics of ecosystems”, li. 116) Ecologists, possibly the main readers of this article, might not be used to the term, information dynamics. Please rephrase it to help their ecological interpretations.

Thanks a lot for this comment. We agree with the reviewer’s suggestion. Before we introduce the optimal information flow (OIF) model, the concept of information dynamics is confusing. Thus, we replaced this expression with “structure and function of fish ecosystems” in section 1.2.

#2 (“optimal information flow model”, li. 135) What is OIF model? It should be explained for non-specialist readers. To my understanding, it is relevant, but does not directly represents interspecific flows of energy or materials, which most ecologists are interested in. It would be helpful if the authors explain what “directed flow of information” means in the context of ecology. Probably, the authors might relate it with “causality” for better understanding by general readers. An explanation that might help readers is in Discussion (“predictive interaction networks”, li. 849).

We fully agree with the reviewer’s concern. The paper was unclear about what the OIF model is with respect to ecological understanding. We added a general explanation for OIF model in the section of “Optimal Information Flow Model and Multi-scale Ecosystem Analysis” in Introduction.

“OIF is based on transfer entropy (TE) with time delays that provides an asymmetric approach to measure directed information flow between random variables (species) from time-series data (Schreiber, 2000; Yao and Li, 2020) and associates cause-effect relationships with directed information flow considering the extent to which one species improves the prediction of another species' future states in the context of the history of another species (Sipahi and Porfiri, 2020). Information flows may relate to different types of interactions, for instance to the collective behavior of species motion (considering data of relative position and alignment between fish species) (Crosato et al., 2018; Brown et al., 2020), or dynamical fluctuations of abundance as in this study.”

We also included specific understanding for the OIF model and directed information flow in ecological senses in Discussion.

“In this study, OIF-inferred networks are predictive interaction networks. They can be used, given source species, to measure the uncertainty reduction of target species. Cause-effect relationships between two species are asymmetrical and the asymmetry is captured by the model.”

#3 It would be essential for general readers to capture the relevance of “information flow” to any ecological processes. The author would explain what it would ecologically mean that Sp. A has a “larger” information flow to Sp. B than to Sp. C. Can we interpret it as Sp. A density has larger per-capita impact on Sp. B than on Sp. B? Does it mean that energy/material flow from Sp. A to Sp. B is larger than flow to Sp. C? I guess not.

Thanks a lot for this comment. We agree with the reviewer. Information theory is a subject in physics and communication technology. Therefore, we fully understand that it is necessary to explain “information flow” in relation to ecological understanding in this paper. 

We added the following content in Discussion (paragraph 2): 

“In this study, time-series observations of species abundance, as research materials, are used by OIF to infer causal networks for the fish community. Information flow is therefore interpreted as the transfer of information about species abundance between species. The amount of the information flow quantifies how much one species can help predict another species in terms of species abundance. In addition, the concept of “information flow” can be also used to represent other indicators in ecosystems dependent on research subjects and objectives, for instance, energy transferred between organisms in food webs, material flow through ecosystems, metabolism expenditure and net primary production.”

For instance, considering time series of species abundance and the concept of TE, if species A has a larger information flow to species B than to species C, it means that given the time series of species A and the history of species B and C, the uncertainty of species B would be eliminated more than that of species C. In other words, species A is a variable that can be used to predict species B better than used to predict species C.

#4 The authors may also explain what is expected in their analysis between two species A and B when they are affected by a common environmental factor C (which might not be observed).

Thanks for this comment. We understand that these results (Table 1) were not explained clearly in the last paragraph of Section 3.4. According to Table 1, species 5 ,6 ,7 ,8 and 9 are the species that are most affected by sea temperature. It is interesting that these species are also recognized as the critical species in the fish community by link salience assessment and OTE. We reorganized the section of 3.4 and included the following content in Paragraph 2.

“Additionally, we also calculate TE, MI, species abundance vs. sea temperature in semi-log scale, Pearson correlation coefficient (indicated as cc) and ρ of CCM (Sugihara et al., 2012) between species abundance and sea temperature (Table 1) to identify which species are most affected by sea temperature. Note that the indicator of slope is the exponent of exponential fitting for species abundance vs. sea temperature (see Fig.S4). In table 1, TE from sea temperature to species abundance for species 5, 6 and 7 is higher than other species, yet MI, Pearson correlation coefficient and ρ are overall higher for species 5, 6, 7, 8 and 9. Therefore, species 5, 6, 7, 8 and 9 are considered as the species that are most affected by sea temperature. It is interesting that these species are also recognized as critical species in the fish community by measuring link salience and OTE. We also find that TE values from species abundance to sea temperature are very small. It is understandable in reality since fish species in the ocean do not affect environmental factors.”

#5 (li. 155, li. 232-242) I understood that the community networks were reproduced for the different TR groups and that some structural differences were identified between them. However, how can the authors conclude that those network structural changes were “caused” by temperature changes? It only tells that there might be a seasonal difference. To conclude that it is due to different temperature, the authors should conduct an additional analysis to detect the information flow from temperature to the network structure, I guess.

Thank you very much for the valuable comment and suggestion. We fully understand the reviewer’s concern. As a part of the whole marine ecosystem, fish community is a complex ecosystem. Sea water temperature is not the only one environmental factor that significantly and directly affects the collective behavior and stability of the fish ecosystem.

To prove sea temperature to be a dominant factor affecting the structure of OIF-inferred networks, we inferred dynamical networks over time for each TR group, then calculated the information flow from sea temperature to the real part of the dominant eigenvalue of TE matrices underlying the network structure (see Table 2). We included the following paragraph in the manuscript:

“In this work, the dominant eigenvalue of species interaction (TE) matrix is studied as a representation of network structure and to analyze the stability of the fish ecosystem. According to the above analysis of species interaction spectrum, OIF-inferred networks representing the fish community present different structural patterns for different TR groups. As a part of the complex marine ecosystem, these structural differences may relate to numerous relevant biotic and abiotic factors. To prove sea temperature to be an important factor affecting the structure of OIF-inferred networks, we infer dynamical networks over time for each TR group, then calculate the information flow from sea temperature to the real part of the dominant eigenvalues of dynamical TE matrices underlying the network structure (see Table 2). The information flow from sea temperature to network structure increases from ≤10℃ to 20-25℃ group, then slightly declines. The lower information flows for ≤10℃ and 10-15℃ groups manifest less effect of sea temperature on these TR groups, implying more stable states with higher resilience to the fluctuation of sea temperature compared to other groups. In contrast, 15-20℃ and 20-25℃ groups show high information flows from sea temperature to network structure, implying strong effect of sea temperature on the fish community. This corresponds to the relatively unstable state (metastability) where these groups undergo a significant change in system state and dynamics. The slight decrease in information flow from 15-20℃ and 20-25℃ groups to ≥25℃ group means the reduction in the effect of sea temperature on (the improvement of system stability for) ≥25℃ group. These results extraordinarily conform to the findings from the probabilistic analysis of species interactions for TR groups. Therefore, the change of sea temperature is considered as an important factor that drives structural changes and state shifts of the fish ecosystem.”

#6 TE based methodology would detect an interaction from Sp. A to Sp. B when the former species indirectly affects the latter via another Species C that mediate the two and the detection probability would be higher when the individual links are stronger. Given this, I would speculate that when strengths of interactions are stronger, the method would detect more interactions from the same system, even if the number of link is in reality not changed. Is it right? Is there possibility that the pattern you found is explained by this?

Thank you very much for the interesting comment. We fully agree with the reviewer’s understanding. This indirect interaction is very likely to happen theoretically. In this paper, even though the OIF model can do the removal of indirect links, we considered all inferred species interactions without checking the redundant information or removing indirect links since we wanted to characterize the full species interactions without any assumptions on biological interactions or methodological criteria of subordinate interactions. This is particularly important when no knowledge is available a priori about biological interactions, whether the biomarker used (e.g. abundance) is reflective of the interactions of interest (physical, biomass conversion, hormonal interactions, etc.), and when indirect weak interactions are quite important (and this is quite common for small organisms, e.g. microbes) (Li and Convertino, 2021).

#7 Ushio et al. (2018) is basically cited only as their data resource, However, the central topic of their study (“to investigate ecosystem stability by inferring species interactions and reconstructing networks”, li. 202-203) is actually the very topic Ushio et al. (2018) dealt with. I think the authors should refer the paper in a more correct way.

Thanks a lot for this comment. We agree with the reviewer. We included the central topic of Ushio et al. (2018) and other relevant studies as a brief literature review on studying the stability of ecosystems using mathematical models in section 1.2. 

“Even though some studies have investigated the stability of ecosystem using mathematical models (Gravel et al., 2016; Ushio et al., 2018; Stone, 2018; May, 2019), integrated methodologies that incorporate network-inferred interaction information and predictive models for understanding marine fish ecosystems are still lacking.”

We also added the citation for the content related to eigenvalues and system stability in Section 2.5 and Discussion (Paragraph 4).

#8 (“Ignoring rare species does not significantly change the marine fish ecosystem and results of this work”, li. 227-228) I wondered why the authors can conclude without analysing the data with more species included.

Thanks for this comment. In short, predictions of other species interactions would not be affected by rare species because these contain very little information in common species, as already found by the study below. Rare species would provide small interactions and yet have little effects on the pdf of interactions and network topology, provided they are not strongly interacting species such as predatory invasive species that sporadically enter the system. Certainly, there is no model that can account for these sporadic events unless they are artificially introduced in the analysis, but that would introduce a set of assumptions that cannot be verified. The sampling effort of the Maizuru Bay is highly reliable and published in top-profile journals.

We clarified this aspect in the second paragraph of section 2.1 as below: 

“since abundance dynamics of rare species contained within other species is typically very small, the inclusion of rare species does not alter the main ecosystem interaction topology beyond just adding poorly interacting species as found in Ushio et al, (2018) and Li and Convertino (2021).” 

Ushio, M., Hsieh, Ch., Masuda, R. et al. Fluctuating interaction network and time-varying stability of a natural fish community. Nature 554, 360–363 (2018).

Li, J., Convertino, M. Inferring ecosystem networks as information flows. Sci Rep 11, 7094 (2021). 

#9 (“native and invasive species groups”, li. 283-285) The authors categorized Plotosus lineatus, Siganus fuscescens and Rudarius ercodes as invasive species with no statements about how it was judged. The author should make clear based on which data or references they were categorized. The same is true for “Fish Stock” categorization. (li. 378) The author should explain what is a rationale of utilizing eigenvalues of TE matrix (not Jacobian matrix based on dynamics of biomass or population densities) as an indicator of system’s stability. None of the studies referred in the MS (Coyte et al., 2015; Ushio et al., 2018; Stone, 2018) do not use TE matrix to evaluate community stability. How relevant is TE-based stability to population dynamical stability?

Thanks a lot for this interesting comment.

We categorized these species into native, invasive and fish stocks groups according to the website of fishbase that is a global fish species database. Take Aurelia aurita as an example, please refer to: https://www.sealifebase.ca/summary/Aurelia-aurita.html. As for this point, we included the link of the “fishbase” website (http://www.fishbase.org/ ) in the paper (Section 2.3).

We fully understand the reviewer’s concern about system stability analyses. Interaction matrix extracted from time series of species abundance not only captures the structure of complex networks, but reflects the dynamics of species abundance and richness themselves. Therefore, interaction matrix describing the connectance of dynamical systems is always used to analyze ecosystem stability. In this study, interaction matrix is inferred by OIF model as TE matrix. We rewrote the beginning of section 2.4.3 relevant to system stability and eigenvalues to explain why we used species interaction matrix to infer system stability and added some citations as follows:

“Interaction matrix defining the connectance of dynamical systems is always used to analyze ecosystem stability considering the whole dynamics and complexity (Gardner and Ashby, 1970; May, 1972; Coyte et al., 2015; Ushio et al., 2018; Stone, 2018. It also defines the topological characterization for dynamical ecosystems, allowing us to analyze them in a systemic view. Furthermore, interaction matrix extracted from time-series data of species abundance captures the structural information of complex ecosystems beyond the dynamics of species abundance and richness themselves. In this study, species interactions are inferred by OIF model as TE. Therefore, we study the ecosystem stability by analyzing eigenvalues of the TE matrix and its distribution in the complex plane.”

“Eigenvalues and eigenvectors provide essential information for reflecting the stability of dynamical systems. Generally, differential equations are used to mathematically describe system dynamics based on variables in the particular system, and eigenvalues and eigenvectors can be used as a method to solve differential equations. The real part of eigenvalues determines whether the system is stable, and the imaginary part determines whether the oscillation is damped or not. If the real part of eigenvalues is negative, the system is stable and damped if the imaginary part is also negative, while undamped if the imaginary part is positive. If the real part of eigenvalues is positive, the system is unstable (Woolf, 2009).”

Gardner, Mark R., and W. Ross Ashby. "Connectance of large dynamic (cybernetic) systems: critical values for stability." Nature 228.5273 (1970): 784-784.

May, Robert M. "Will a large complex system be stable?." Nature 238.5364 (1972): 413-414.

#10 (“This result means that increasing fluctuations of sea temperature lead to biodiversity loss for the fish community”, li. 451-452) There might be a correlation, but no reason to believe there is a causation between them. Such a speculation should be removed from Result section.

Thanks for this comment. We totally agree with the reviewer. We modified the relevant contents in sections of Results, Discussion and Conclusions, as follows:

“This result implies that the change of biodiversity loss in the fish community may relate to the increasing fluctuations of sea temperature.” 

“These analyses capture the seasonal fluctuations of sea temperature, species populations and diversity, and the results support a speculation that the change of biodiversity in the fish community can be a biological response to the increasing fluctuations of sea temperature.”

“Accordingly, the biodiversity loss in the fish community may relate to the increasing fluctuations of sea temperature.”

#11 (“The highest total abundance of species in EP, FS and native groups in 2009 is likely owing to the least temperature fluctuation in this year”, li. 457-458) Again, any sentences with speculation and interpretation in Result should be moved to Discussion. The same applies to many other parts (for example, “This finding means…” (li. 536-540)) and the main text seems to need extensive re-constructions.

Thanks for this valuable comment. We fully understand the reviewer’s concern. We moved the contents of speculation and interpretation in Results to Discussion, and reorganized the section of Discussion as follows:

Line 457-458, 466-469 in Results to paragraph 1 in Discussion:

“These analyses capture the seasonal fluctuations of sea temperature, species populations and diversity, and the results support a speculation that the change of biodiversity in the fish community can be a biological response to the increasing fluctuations of sea temperature. The species abundance of fish groups shows an overall slight increase since 2007 that is related to sea temperature. Take 2009 as an example, the highest total abundance of species in EP, FS and native groups in this year is likely owing to the least fluctuation of sea temperature. The increase of species abundance may improve species competition in the fish community. Those species whose abundances decrease with the increasing sea temperature are supposed to compete with other species and present disadvantage in species interactions. This competitive disadvantage may lead to departure and extinction of some species, resulting in the global decrease of species diversity in the fish community.”

Line 536-540 in Results to paragraph 4 in Discussion:

“This finding means that species 6, 7, 8 and 9 are core species in the fish community that are interacting strongly with each other, present quite different behavior compared to other species, and play a vital role in maintaining the stability of the ecosystem. More importantly, these species are native species in Maizuru bay (see Table. S1).”

Line 581-584, 612-614 in Results to paragraph 3 in Discussion:

“For ≤10℃ and 10-15℃ groups, the fish ecosystem lies on a globally stable state, then presents a shift of system state from stability to metastability in intermediate temperature ranges, and finally returns to a locally stable state. Fish community on global stability is in the situation that the fish ecosystem is highly resistant to long-term perturbations (change of species composition, sea temperature, for instance) (Nunney, 1980; Chen and Cohen, 2001), and intra- and inter-specific interactions are not too strong (Goh, 1977). The newly established stable state is interpreted as locally stable state which addresses the fish ecosystem only resistant to small short-lived disturbances (minor fluctuation of sea temperature, for instance) (Ak, Gumus, 2014).”

Line 709-711 in Results to paragraph 4 in Discussion:

“It is intuitively speculated that the species whose abundances are more uniformly distributed play a cooperative role in the fish community that is beneficial to the increase of abundance, but do not have strong effects on other species.”

#12 (“Eigenvalues and eigenvectors…(Woolf, 2009)”, li. 625-633) This part explains the methodology and should be moved to Materials & Methods.

Yes, we fully agree with the reviewer’s comment. We moved this part to the first paragraph of section 2.4.3. Thanks a lot for noticing it.

Minor Points

#13 (“seafood web”, li. 105) Probably “sea food web”?

Yes. We found the spelling mistake, and changed it to “marine food webs” that seems a more commonly used description. Thanks for noticing it.

#14 (“Purely biological analyses”, li. 125) What does it mean? Ecology is not biology here?

Thank you very much for this valuable comment. 

“Purely biological analyses” was used to describe the analyses that only consider species abundance, composition and richness. We understand that this expression is unclear and misleading. We don’t use it in the paper any more. 

#15 (li. 328) Not “Schreiber (2000)” , but “(Schreiber 2000)” here.

Yes, we corrected it. Thanks a lot for noticing it.

#16 (“we exploit eigenvalues of TE matrix as an indicator to evaluate network stability”,

Thanks for noticing it. Yes, we revised the English of the sentence as below: 

“we exploit eigenvalues of the TE matrix as indicators of network stability”

#17 (“Pseudolabrus.sieboldi”, li. 838) Remove “.” Between “Pseudolabrus” and “sieboldi”.

Yes. We corrected Table 1, S1 and S2 and other species names in the paper. Thanks for noticing it.

---

## [Editor Report · Decision Letter 1]

14 Sep 2021

Temperature Increase Drives Critical Slowing Down of Fish Ecosystems

PONE-D-21-01412R1

Dear Dr. Convertino,

We’re pleased to inform you that your manuscript has been judged scientifically suitable for publication and will be formally accepted for publication once it meets all outstanding technical requirements.

Kind regards,

Andrea Belgrano, Ph.D.

Academic Editor

PLOS ONE

Additional Editor Comments (optional):

The revised manuscript fully addressed the comments and suggestions made by reviewers #1 and #2.  In particular I found very interesting the reconstruction of species interaction networks and the different topologies that emerge in relation to both temperature changes and species dynamics .

---

## [Editor Report · Acceptance letter]

22 Sep 2021

PONE-D-21-01412R1 

Temperature Increase Drives Critical Slowing Down of Fish Ecosystems 

Dear Dr. Convertino:

I'm pleased to inform you that your manuscript has been deemed suitable for publication in PLOS ONE. Congratulations! Your manuscript is now with our production department. 

Kind regards, 

on behalf of

Dr. Andrea Belgrano 

Academic Editor

PLOS ONE